# Leveraging a Novel Hybrid Ensemble and Optimal Interpolation Approach for Enhanced Streamflow and Flood Prediction

Mohamad El Gharamti[1], Arezoo RafieeiNasab[2], and James L. McCreight[3]

[1]NCAR, Computational and Information Systems Laboratory (CISL), Boulder CO, USA
[2]NCAR, Research Applications Laboratory (RAL), Boulder CO, USA
[3]UCAR, CPAESS Cooperative Programs for the Advancement of Earth System Science, Boulder CO, USA

**Correspondence:** Moha Gharamti (gharamti@ucar.edu)

**Abstract.** In the face of escalating instances of inland and flash flooding spurred by intense rainfall and hurricanes, the accurate prediction of rapid streamflow variations has become imperative. Traditional data assimilation methods face challenges during extreme rainfall events due to numerous sources of error, including structural and parametric model uncertainties, forcing biases and noisy observations. This study introduces a cutting-edge hybrid ensemble and optimal interpolation data assimilation scheme tailored to precisely and efficiently estimate streamflow during such critical events. Our hybrid scheme uses an ensemble-based framework, integrating the flow-dependent background streamflow covariance with a climatological error covariance derived from historical model simulations. The dynamic interplay (weight) between the static background covariance and the evolving ensemble is adaptively computed both spatially and temporally. By coupling the National Water Model (NWM) configuration of the WRF-Hydro modeling system with the Data Assimilation Research Testbed (DART), we evaluate the performance of our hybrid prediction system using two impactful case studies: 1. West Virginia's flash flooding event in June 2016, and 2. Florida's inland flooding during Hurricane Ian in September 2022. Our findings reveal that the hybrid scheme substantially outperforms its ensemble counterpart, delivering enhanced streamflow estimates for both low and high flow scenarios, with an improvement of up to 50%. This heightened accuracy is attributed to the climatological background covariance, mitigating bias and augmenting ensemble variability. The adaptive nature of the hybrid algorithm ensures reliability even with a very small time-varying ensemble. Moreover, this innovative hybrid data assimilation system propels streamflow forecasts up to 18 hours in advance of flood peaks, marking a substantial advancement in flood prediction capabilities.

## 1 Introduction

Flooding can stem from various causes, including prolonged rainfall events like tropical storms or hurricanes, as well as intense rainfall over short periods or complications such as debris and ice jams. When examining events causing at least a billion dollars in damage, river and urban flooding alone account for 7.4% of US natural disasters from 1980 to 2023. Tropical cyclones top the list, contributing to 52% of the damage (Smith, 2020).

Tropical storms and hurricanes are characterized by destructive winds, storm surge, and catastrophic flooding. Hurricanes can unleash 2.4 trillion gallons of rainwater in a day (Nunez and Yang, 2023). According to the National Weather Service, torrential rain from hurricanes can flood the neighborhoods of coastal communities within minutes. This phenomenon, known

as freshwater or inland flooding, can damage infrastructure, cause landslides, destroy crops, and take lives. Inland flooding could be caused by both the river water level exceeding river bank heights or rainfall intensity exceeding the infiltration capacity. The latter is the major cause of the flooding in case of tropical storms and hurricanes. Approximately, 25% of US hurricane deaths from 1963 to 2012 were related to freshwater flooding (Rappaport, 2014). Predicting floods has the potential to save lives, protect infrastructure, and minimize socio-economic impacts. Such predictions are challenging and remain an area of active research and operational development.

Streamflow predictions commonly integrate real-world observations with hydrologic model simulations, a practice known as data assimilation (DA) (Houser et al., 1998; Margulis et al., 2002; Reichle, 2008; Yucel et al., 2015). This approach, prevalent across diverse fields including numerical weather prediction (NWP; Lorenc, 1986), has witnessed substantial growth in streamflow prediction applications over the last two decades (e.g., Kim et al., 2021; Chao et al., 2022). Ensemble filtering techniques, based on the Kalman filter (Kalman, 1960), are widely adopted due to their ease of implementation and high portability. The ensemble Kalman filter (EnKF; Evensen, 1994; Burgers et al., 1998) stands out among these methods, utilizing model forecast realizations during the analysis. The EnKF employs a minimum variance estimator, utilizing observations to compute ensemble increments (i.e., difference between analysis and forecast) that are subsequently linearly combined with the predicted ensemble. Extensively used in real-time and operational settings, the EnKF has been applied to enhance streamflow simulations. For instance, Pauwels and De Lannoy (2006, 2009) investigated a retrospective EnKF formulation in several conceptual and operational rainfall-runoff models and assessed the impact of streamflow DA using linear and nonlinear observation operators. Rakovec et al. (2012) explored performance of the EnKF in updating streamflow through synthetic and real-world experiments in the Meuse River basin, Belgium. In an operational hydrological context, McMillan et al. (2013) introduced a recursive EnKF variant stabilizing streamflow estimates across different catchments in New Zealand. Rafieeinasab et al. (2014) compared the performance of the maximum likelihood ensemble filter with the EnKF for real-time streamflow assimilation in a Southern Texas headwater basin. Meanwhile, Huang et al. (2017) utilized the EnKF to update streamflow using snow water equivalent data in basins like the Pacific Northwest, Rocky Mountains, and California in the western US. Additionally, Lee et al. (2019) developed a bias-aware version of the EnKF to enhance flood forecasting for a subset of 10 headwater basins in the Southern US. Beyond the EnKF, various methods, including particle filters (e.g., Weerts and El Serafy, 2006; DeChant and Moradkhani, 2011; Noh et al., 2013; DeChant and Moradkhani, 2014; Abbaszadeh et al., 2020), ensemble smoothers (e.g., Margulis et al., 2015; Meng et al., 2017), variational methods (e.g., Seo et al., 2009; Mazrooei et al., 2021), and machine learning (e.g., Boucher et al., 2020), have been employed for streamflow and flood prediction across diverse spatial and temporal scales. A comprehensive review of methods used in streamflow prediction over the past four decades is available in Troin et al. (2021).

Despite its achievements in research, real-time applications, and operational frameworks, the EnKF operates as a sequential estimation tool with inherent limitations, grappling with biases and sampling errors. The EnKF approximates the true prior (or forecast) covariance by employing a sample covariance from the ensemble, assuming unbiased background errors. In scenarios with small ensemble sizes, the estimated sample covariance may be plagued by substantial sampling errors, leading to suboptimal analysis (or posterior) estimates. These errors often result in the underestimation of the true ensemble variance, po-

tentially causing filter divergence in severe situations (Furrer and Bengtsson, 2007). Moreover, the use of a restricted ensemble size can introduce spurious correlations in space, proving detrimental over successive assimilation cycles (Anderson, 2012). Model biases, typically unaccounted for in the EnKF, tend to dominate other error sources, potentially causing catastrophic consequences such as complete ensemble collapse (Sacher and Bartello, 2008). Localization and inflation are commonly employed methods to address sampling errors. Localization (Houtekamer and Mitchell, 2001) restricts the impact of observations to nearby state variables, mitigating nonphysical correlations, especially between spatially distant observations and state variables. Inflation (Anderson and Anderson, 1999) increases the ensemble spread around its mean, countering sampling errors and enhancing the fit to observations. Both inflation and localization have become integral tools in the majority of streamflow ensemble DA studies in the literature (e.g., Emery et al., 2020). Additional techniques addressing sampling errors encompass relaxation of prior spread and perturbations (Zhang et al., 2004; Whitaker and Hamill, 2012), the utilization of stochastic perturbations and multi-physics ensembles (e.g., Berner et al., 2011), and covariance hybridization (Hamill and Snyder, 2000). Beyond sampling errors and biases, issues such as non-linearity and non-Gaussianity frequently compromise the optimality of the EnKF update step (Anderson, 2010).

Integrating the EnKF with other DA methods can mitigate some of its shortcomings. For instance, optimal interpolation (OI) or three-dimensional variational (3D-Var) systems[1] avoid sampling errors by depending on a time-invariant background error covariance, typically estimated from the model's climatology. This approach has been extensively explored in atmospheric and ocean DA literature (Counillon et al., 2009; Bannister, 2017, and references therein). However, in the realm of streamflow applications, the utilization of hybrid ensemble and variational DA techniques remains limited and is actively researched. In a study on high-resolution hydrologic forecasting, Hernández and Liang (2018) investigated streamflow predictive accuracy using a hybrid scheme that combines particle filtering (PF) with four-dimensional variational (4D-Var) data assimilation. The authors reported the hybrid algorithm's ability to provide skillful streamflow predictions, accommodating non-Gaussian, non-linear, and high-resolution estimation. In another exploration, Abbaszadeh et al. (2019) developed a similar hybrid PF and 4D-Var system, assessing its performance across several river basins in the US. Their hybrid scheme, adept at handling various sources of uncertainties, proved efficient and robust to the number of particles. In this work, we delve into the functionality, performance, and efficiency of a hybrid EnKF and OI scheme, hereafter referred to as EnKF-OI, in the context of flash flooding and freshwater events. We meticulously examine the weighting between the ensemble-based flow-dependent covariance and the climatological static background covariance. Furthermore, we implement the adaptive hybrid weighting scheme proposed by El Gharamti (2021) and investigate the temporal and spatial variations of the weighting factor across the stream network. Our analysis extends to assessing the impacts of the hybrid scheme on streamflow biases and sampling errors. To our knowledge, this marks the inaugural application of an adaptive hybrid EnKF and 3D-Var DA scheme to real-world flood forecasting problems.

The data assimilation framework employed in this study is based on the integrated HydroDART system, as detailed by El Gharamti et al. (2021). This system utilizes NOAA's National Water Model (NWM) configuration within the WRF-Hydro

---

[1]Throughout this study, the terms OI and 3D-Var are used interchangeably, both solving the same DA problem and relying on the same static background error covariance matrix.

hydrological framework (Gochis et al., 2020). Our implementation involves a sub-model configuration of the NWM, specifically designed to incorporate both streamflow and conceptual groundwater storage. To facilitate the assimilation process, the model is interfaced with the Data Assimilation Research Testbed (DART, Anderson et al., 2009). The HydroDART system, as highlighted in El Gharamti et al. (2021), incorporates Along-The-Stream (ATS) localization, along with adaptive prior and posterior inflation, to enhance ensemble performance. To extend the capabilities of HydroDART, we introduce a hybrid method using a 42-year retrospective run of WRF-Hydro covering the entire Contiguous United States (CONUS). This extensive model run is leveraged to construct climatological error covariances for both streamflow and groundwater bucket storage. The implementation of the hybrid ensemble-variational scheme is the first of its kind within DART and features several flavors for updating the hybrid weighting coefficients including: constant weights, time-varying homogeneous weights in addition to the more comprehensive temporally and spatially varying weights (as in this work).

Our EnKF-OI prediction system undergoes testing in two regional flooding scenarios. The initial case, spanning June 2016, addresses an 11-day flash flooding event in West Virginia. The second case study focuses on inland flooding caused by Hurricane Ian in central Florida (from September 15th to October 15th, 2022). In both instances, streamflow observations obtained from United States Geological Survey (USGS) gauges are assimilated at hourly time steps. Our evaluation encompasses an assessment of the performance of both hybrid (EnKF-OI) and non-hybrid (EnKF) DA methodologies in improving simulated hydrographs under diverse flooding conditions. To gain insights into the behavior of the hybrid EnKF-OI algorithm, we conduct sensitivity experiments concerning the hybrid weighting coefficient and the ensemble size. Furthermore, the analyses derived from the hybrid scheme's posterior states contribute to generating short-range streamflow forecasts, allowing us to evaluate the impact of data assimilation on these predictions.

The subsequent sections delineate the paper's organization. Section 2 delves into the detailed description of the integrated HydroDART prediction system, shedding light on both the model and the DA tool. Particular attention is paid to elucidating the methodology employed in generating the static-background covariance. Section 3 expounds on the specifics of the test cases, providing insights into the hydrologic domains' extent and elaborating on the USGS observations. Moving forward, Section 4 unfolds the results obtained from the EnKF and the hybrid EnKF-OI methodologies across the two hydrologic domains. Distinct spatial and temporal evaluations underscore the hybrid algorithm's performance. The concluding insights and broader discussions emanate in Section 5, encapsulating a comprehensive summary of the findings.

## 2 Model and Methods

### 2.1 WRF-Hydro

The Weather Research and Forecasting, Hydrological modeling system (WRF-Hydro) is used widely across the hydrology community both in coupled and uncoupled modes with atmospheric models (e.g., Senatore et al., 2015; Kerandi et al., 2018; Wang et al., 2022; Son et al., 2023). A prominent application of WRF-Hydro is the National Water Model (NWM) which became operational in August 2016 and has gone through several version upgrades since then. The National Water Model is a particular configuration of the uncoupled WRF-Hydro system which is operational over CONUS, Hawaii, Puerto Rico

and Virgin Islands (NWMv2.1) and Alaska (NWMv3.0). The modules and physics in this paper are equivalent to the NWM version 2.1 standard analysis and assimilation cycle without the streamflow nudging used in the NWM. Streamflow nudging is the current data assimilation methodology in NWM operationally. "Nudging" also known as direct insertion refers to moving the modeled flow towards the observed discharge at each time step of the routing model.

The NWMv2.1 configuration consists of the Noah-MP land surface model (Niu et al., 2011; Yang et al., 2011), subsurface and surface flow routing, baseflow/groundwater routing, channel and lake/reservoir (i.e. waterbodies) routing (Cosgrove et al., 2024). In each time step, first the land surface model is operated on a coarse resolution of 1 km$^2$. Then, terrain routing (subsurface and surface flow routing) is performed on the 250 m$^2$ grid spacing. NWM utilizes the USGS National Hydrography Data (NHD) Plus Version 2 medium-resolution dataset (McKay et al., 2012), which provides both streams and corresponding

catchments. Each stream is represented by a channel/reach vector in the model, and the basin associated with the stream acts as a conceptual groundwater basin/bucket in the model. The inflow to each groundwater bucket/basin is the aggregated outflow from the soil column (1 km land surface model grid) to the NHDPlusV2 catchments. Then, a conceptual groundwater routing is performed. The outflow from bucket/basin is estimated based on an exponential storage-discharge function. Next, the outflow from groundwater basin/bucket combined with the lateral channel inflows from the terrain routing are routed through

the channels using the Muskingum-Cunge routing method (Read et al., 2023). WRF-Hydro also includes options to represent lakes and reservoirs. Inflow fluxes to lakes and reservoir objects embedded into the NWM routing network are routed using a level pool scheme (Gochis et al., 2020; Cosgrove et al., 2024).

We use a channel, reservoir, and conceptual groundwater submodel of the NWM is used, following El Gharamti et al. (2021). This configuration is computationally cheaper compared to the full model and therefore appealing for running an ensemble

system. The prognostic variables that are updated in this study are the streamflow discharge and groundwater bucket head. It should be noted that the lake/reservoir objects are defined on the stream reaches, however they are not considered in the state updating. Figure 1 shows the chain of modeling components in our system and how the deterministic fluxes from the full NWM model arrive as boundary conditions to the HydroDART system.

Random noise is applied to these deterministic boundary fluxes to generate an ensemble of input fluxes for the streamflow,

reservoir, and conceptual groundwater system. In addition to this time-varying uncertainty, we also generate an invariant ensemble of channel parameters similar to the configuration of El Gharamti et al. (2021). These two levels of perturbations were found necessary to generate larger variability in the predicted ensemble. Because ensemble DA depends on probabilistic forecasts, enhancing the variability within the ensemble can aid the method in accurately estimating the states of the hydrological model.

To perturb the boundary fluxes to the streamflow and bucket models, we use Gaussian samples with zero mean and standard deviation equal to 40% of the flux value at each location. The Gaussian choice yielded the best streamflow estimates (in terms of accuracy and spread consistency) as compared to other tested distributions (e.g., gamma, inverse-gamma and exponential). We also perturb the geometric and other channel parameters using uniform noise models. The parameters of the uniform densities were carefully selected such that the resulting streamflow ensembles were found not only skillful but also reliable.

For detailed description of the WRF-Hydro configuration used in this paper as well as the creation of the forcing ensemble

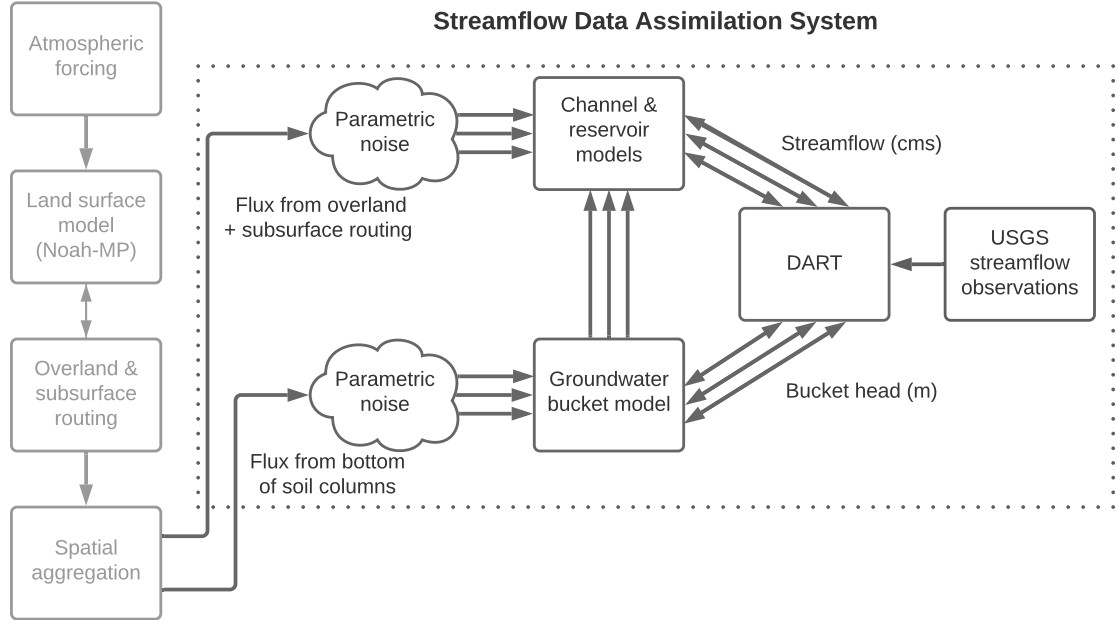

**Figure 1.** Streamflow data assimilation workflow adapted from El Gharamti et al. (2021). The vertical boxes on the left depict the WRF-Hydro components that are executed once and produce the deterministic fluxes into the channels and groundwater buckets. The arrows represent the order of the physics component execution and a two way arrow represents two way coupling. The dotted box represents the HydroDART components. Random noise is applied to the deterministic input fluxes to HydroDART generating an ensemble of input fluxes for both channel and conceptual groundwater models. Three arrows denote the presence of ensembles, however, the ensemble size is larger than three. As shown in this workflow, DART assimilates USGS streamflow observations and updates the streamflow (in cubic meters per second) and bucket head (in meters) state variables in the channel and reservoir model, and groundwater bucket model, respectively.

and channel parameter ensemble, please refer to El Gharamti et al. (2021). Here, the model code (https://github.com/NCAR/wrf_hydro_nwm_public/releases/tag/nwm-v2.1-beta3), domain data and parameter sets (https://www.nco.ncep.noaa.gov/pmb/codes/nwprod/nwm.v2.2.3/parm/domain/) are based on the NWMv2.1. There are a number of parameters in the WRF-Hydro modules, in particular, the NoahMP land surface model (Mendoza et al., 2015; Cuntz et al., 2016; He et al., 2023), with different degrees of sensitivity that could be tuned via calibration. In NWMv2.1, 14 parameters impacting different processes (vegetation, soil, snow, and runoff parameters) were chosen based on the previous sensitivity analysis and studies (Cosgrove et al., 2024). These parameters were calibrated using the iterative Dynamically Dimensioned Search approach (Tolson and Shoemaker, 2007) for a large number of basins throughout the US. The objective function[2] was one minus weighted Nash-Sutcliffe efficiency (NSE, Nash and Sutcliffe, 1970) and NSE of logarithmic streamflow (NSElog), both calculated based on

---
[2]Minimize $J = 1 - 0.5\,(\text{NSE} + \text{NSElog})$.

the hourly streamflow simulations. Summary model statistics prior and after calibration can be found in Cosgrove et al. (2024). It should be noted that some model biases remain after the calibration process.

## 2.2 DART

### 2.2.1 Ensemble Filtering

The Data Assimilation Research Testbed (DART, Anderson et al., 2009) is a sequential ensemble DA tool developed and maintained at the National Center for Atmospheric Research. DART employs different variants of the ensemble Kalman filter for linear and nonlinear estimation problems. The filtering schemes are based on Bayes rule such that the prior distribution is recursively updated using the observation likelihood to obtain a posterior probability density function (pdf). For HydroDART, several streamflow realizations are first integrated forward in time using the hydrologic model, WRF-Hydro, denoted by $\mathcal{M}$, until the next observations become available.

$$\mathbf{x}_k^{f,i} = \mathcal{M}\left(\mathbf{x}_{k-1}^{a,i}, \boldsymbol{\theta}^i, \boldsymbol{\gamma}_k^i\right), \qquad i = 1, 2, \ldots, N_e \tag{1}$$

Here, $\mathbf{x}$ denotes the hydrologic state which consists of streamflow and the conceptual groundwater storage. The superscripts $f, a, i$ denote forecast, analysis and ensemble member, respectively. The subscript $k$ denotes time and $N_e$ is the ensemble size. $\boldsymbol{\theta}$ is a set of 6 physical parameters that describe the geometry of the streamflow compound channel. The parameters are: top channel width, bottom width, side slope, Manning's N, width of the compound channel and Manning's N of the compound channel. Notice that for each member, a different configuration of these channel parameters is used as outlined in El Gharamti et al. (2021). Perturbed boundary fluxes to the streamflow and the groundwater models are given in $\boldsymbol{\gamma}$. Using equation (1), the first and second moments of the prior distribution are approximated as follows:

$$\overline{\mathbf{x}}_k^f = \frac{1}{N_e} \sum_{i=1}^{N_e} \mathbf{x}_k^{f,i}, \tag{2}$$

$$\mathbf{P}_k^f = \lambda \cdot \overline{\mathbf{P}}_k^f, \tag{3}$$

$$= \lambda \cdot \frac{1}{N_e - 1} \sum_{i=1}^{N_e} \left(\mathbf{x}_k^{f,i} - \overline{\mathbf{x}}_k^f\right) \left(\mathbf{x}_k^{f,i} - \overline{\mathbf{x}}_k^f\right)^T, \tag{4}$$

where $\overline{\mathbf{x}}_k^f$ and $\overline{\mathbf{P}}_k^f$ are the prior ensemble mean and the prior sample covariance, respectively. The coefficient $\lambda$ in equation (4) is an inflation factor used to restore the ensemble variance after the forecast. In this study, the inflation factor is selected adaptively in space and time using the adaptive inflation scheme of El Gharamti (2018). We also apply adaptive posterior inflation to the analysis ensemble according to El Gharamti et al. (2019). The estimated (inflated) prior covariance at time $t_k$ is given by $\mathbf{P}_k^f$.

At the time of the analysis, DART assimilates the observations serially according to Anderson (2003):

$$\Delta y^i = y^{a,i} - y^{f,i}, \tag{5}$$

$$\mathbf{x}_{j,k}^{a,i} = \mathbf{x}_{j,k}^{f,i} + \rho \frac{\sigma_{xy}}{\sigma_y^2} \Delta y^i, \qquad j = 1, 2, \ldots, N_x. \tag{6}$$

The subscript $j$ is an index to the variables in the state and the total number of state variables is denoted by $N_x$. $\sigma_{xy}$ is the prior covariance between the observation $y$ and $j^{\text{th}}$ state element while $\sigma_y^2$ is the sample variance of the observed variable. As can be shown, the EnKF solution is obtained as a linear regression of the observation increments $\Delta y$ on the entire state vector. We note that the assimilated observations are noisy with Gaussian errors and their observation error covariance is accounted for when computing the observation increments in (5). $\rho$ is a localization coefficient typically between 0 and 1 and is computed using the common Gaspari-Cohn correlation function (Gaspari and Cohn, 1999). The localization strategy employed here follows the ATS localization scheme (El Gharamti et al., 2021) such that only reaches upstream and downstream from a particular gauge are updated during the analysis. In terms of implementation, we note that the full state covariance in eq. (4) is never constructed using this 2-step serial update scheme. This also applies for the observation error covariance matrix assuming that the observations are uncorrelated in space. For more details on the algorithm and its implementation, the reader is referred to the work of Anderson (2003).

### 2.2.2 Hybridized Covariance

Hybridizing the prior ensemble covariance matrix is performed linearly right before the update:

$$\mathbf{P}_k^h = \alpha_k \mathbf{P}_k^f + (1 - \alpha_k)\mathbf{B}, \tag{7}$$

where $h$ denotes the hybrid form of the prior covariance and $\mathbf{B}$ is a static background covariance matrix. $\alpha_k$ is a weighting factor chosen between 0 and 1. Notice that $\mathbf{B}$ is a stationary covariance and is typically used in 3-4D variational systems. $\mathbf{B}$ can be estimated from the model's climatology using a large inventory of historical forecasts. In NWP systems, for instance, the National Meteorological Center (NMC, Parrish and Derber, 1992) technique is often used to compute $\mathbf{B}$. More details on the computation of $\mathbf{B}$ for the current streamflow work can be found in the next section. From equation (7), one may argue that $\mathbf{P}_k^h$ is a better estimate of the true covariance compared to the ensemble one $\mathbf{P}_k^f$. This is because blending in climatological information in the ensemble covariance usually presents the ensemble with new correction directions (Hamill and Snyder, 2000). When $\alpha_k$ is set to 1, equation (7) results in a purely flow-dependent covariance reducing the algorithm to the EnKF. In contrast, when $\alpha_k = 0$ the system morphs into an ensemble optimal interpolation (EnOI) scheme. The hybrid EnKF-OI scheme is activated for $0 < \alpha_k < 1$.

Rather than manually tuning $\alpha_k$, one can adaptively estimate it (El Gharamti, 2021) using Bayes rule:

$$p(\alpha_k|\mathbf{d}_k) \approx p(\alpha_k) \cdot p(\mathbf{d}_k|\alpha_k), \tag{8}$$

where $\mathbf{d}_k$ is the background innovation (i.e., observation minus forecast). Equation (8) assumes that $\alpha_k$ is a random variable with prior distribution $p(\alpha_k)$ which is considered Gaussian in this work. The likelihood term on the right hand side of (8) is also assumed Gaussian with zero mean and covariance:

$$\mathbb{E}\left[\mathbf{d}_k\mathbf{d}_k^T\right] = \mathbf{R}_k + \mathbf{H}_k\mathbf{P}_k^t\mathbf{H}_k^T, \tag{9}$$
$$\approx \mathbf{R}_k + \alpha_k\mathbf{H}_k\mathbf{P}_k^f\mathbf{H}_k^T + (1-\alpha_k)\mathbf{H}_k\mathbf{B}\mathbf{H}_k^T. \tag{10}$$

This result holds as long as the background and observation errors are uncorrelated (Desroziers et al., 2005). $\mathbf{H}_k$ is the observation operator and $\mathbf{R}_k$ is the observation error covariance matrix. The hybridized covariance $\mathbf{P}_k^h$ is assumed to be equivalent to the true background covariance $\mathbf{P}_k^t$ allowing the insertion of equation (7) in (9). Using equation (10), the likelihood of the weighting coefficient can be written as:

$$p\left(\mathbf{d}_k|\alpha_k\right) \quad = \quad \left(\sqrt{2\pi}\beta\right)^{-1} \exp\left[-\frac{\mathbf{d}_k^T \mathbf{d}_k}{2\beta^2}\right], \text{ where} \tag{11}$$

$$\beta \quad = \quad \sqrt{\sum_i (\mathbf{R}_k)_{ii} + \alpha_k \sum_i (\mathbf{H}_k \mathbf{P}_k^f \mathbf{H}_k^T)_{ii} + (1 - \alpha_k) \sum_i (\mathbf{H}_k \mathbf{B} \mathbf{H}_k^T)_{ii}}. \tag{12}$$

The notation $\sum_i (\mathbf{R}_k)_{ii}$ is equivalent to the trace of matrix $\mathbf{R}_k$. Taking the product of the likelihood and the prior of $\alpha_k$ results in a Gaussian posterior which can be used in subsequent DA cycles. The updated value of $\alpha_k$ is obtained by maximizing its posterior $p(\alpha_k|\mathbf{d}_k)$ pdf. Following this formulation, a spatially and temporally adaptive weighting coefficient can be computed. This algorithm has been incorporated within the coupled streamflow prediction system, HydroDART. More details on the adaptive hybrid EnKF-OI scheme can be found in El Gharamti (2021).

## 3 Test Cases

### 3.1 Domains

Two extreme flooding events are selected for this study. One is a flash flood resulting from a thunderstorm. The other is a long lasting flooding event resulting from a hurricane landfall. Figure 2 shows the location and extent of the two domains and they are explained in more detail below.

### 3.1.1 Florida's Flooding Case (2022)

Hurricane Ian became a tropical storm on September 24, 2022. Ian made landfall on western Cuba as a high-end category 3 hurricane on September 27. On September 28, Ian made landfall on southwestern Florida as a category 4 hurricane, producing catastrophic storm surge and historic freshwater flooding across much of central and northern Florida. Ian was responsible for more than 156 fatalities in the United States of which 66 were directly caused by the storm. Ian caused over $112 billion in damages; the costliest hurricane in Florida's history and the third costliest in United States history (National Hurricane Center Tropical Cyclone Report; Bucci et al., 2023).

Figure 2-(a) shows the WRF-Hydro modeling domain for hurricane Ian. The colored map background depicts accumulated rainfall drawn for September 28 through 30, 2022, as modeled by the NWMv2.1 Analysis and Assimilation atmospheric forcing (described below) used to drive this case. The domain is a subset of the NWMv2.1 CONUS domain and includes 18,190 reaches and 151 lakes and reservoirs. Stream reaches are color coded based on their 10-year flood magnitudes as calculated from the 42 year NWMv2.1 respective model simulations described in more detail below. There are 171 USGS gauges with their drainage area fully contained in this domain and are assimilated and also used in performance assessment (dark red circles in Figure 2). Twenty-two of these gauges are GAGES-II reference gauges (Falcone, 2011) which have little or no anthropogenic alterations

to their natural streamflows (green squares). Since WRF-Hydro does not have an active reservoir model and is performing a simple level pool routing, GAGES-II reference gauges are better suited for verification because they avoid accounting for heavy flow regulation at many reservoirs. Figure 2-(a) also shows the location of five labeled verification gauges (black triangles) for which streamflow time-series will be provided in the results (section 4).

The full WRF-Hydro/NWMv2.1 model was run (without nudging data assimilation) for the FL's Ian test case using the NWMv2.1 analysis and assimilation cycle forcing dataset (https://water.noaa.gov/about/nwm). This meteorological forcing set is drawn from the Multi-Radar Multi-Sensor (MRMS) Gauge-adjusted and Radar-only observed precipitation products along with short-range Rapid Refresh (RAP) and High-Resolution Rapid Refresh (HRRR) . These atmospheric forcings drive the single model run depicted vertically on the left side of Fig. 1 and produce the output fluxes used by HydroDART to generate ensemble forcing for the channel+conceptual groundwater submodel. The simulation period is from August 15 to October 15, 2022. The model is initialized based on the operational NWMv2.1 model states on August 15, then the first month is used as a spin up period and streamflow assimilation begins on September 15, 2022.

### 3.1.2 West Virginia's Flooding Case (2016)

Several rounds of thunderstorms on June 23[rd], 2016 produced torrential rainfall in West Virginia (WV) and western-central Virginia. Record rainfall accumulations of 200-250 mm were observed over a 24 hour period ending on 1200 UTC of June 24[th] (Martinaitis et al., 2020). This resulted in a rapid rise of water (in some places less than 6 hours) and extensive flooding across the domain. This was one of the deadliest flooding events in West Virginia's history with 23 fatalities. The event was classified as a billion-dollar disaster which damaged thousands of structures and over 1500 roads and bridges (Martinaitis et al., 2020). Many USGS gauges in the domain reported some level of flooding. Five of those gauges had their record flood stage measured in this event. The data assimilation and verification period for this test case is from June 20, 2016 through June 30, 2016; an 11 day period which encompasses the flash flooding event and the recession period completely at all verification gauges.

Figure 2-(b) shows the WV flooding domain. Just like FL's Ian, the domain is a cutout of the NWMv2.1 CONUS domain including 47,046 reaches and 25 lakes and reservoirs. There are 121 USGS gauges with their drainage area fully contained in the modeling domain. These gauges are assimilated and used in performance assessment. Figure 2-(b) shows the location of all those gauges as well as the subset of GAGES-II reference gauges. Figure 2-(b) also shows the location of a few verification gauges for which streamflow time series are provided and analyzed in section 4.

Analysis of Record for Calibration (AORC, Fall et al., 2023) is used as the atmospheric forcing for the West Virginia test case. This is a new dataset developed to support NWM calibration and long term retrospective model simulations. AORC data has been cut out to the West Virginia modeling domain and used to force the WRF-Hydro model. The inflows to the channel and conceptual groundwater from this simulation are then used as forcing in all experiments (both data assimilation and forecasts). The simulation period is from June 1 through June 31, 2016. The model is initialized based on the NWMv2.1 retrospective model states on June 1, then the first 20 days are used as a spin up period and streamflow assimilation begins on June 20, 2016.

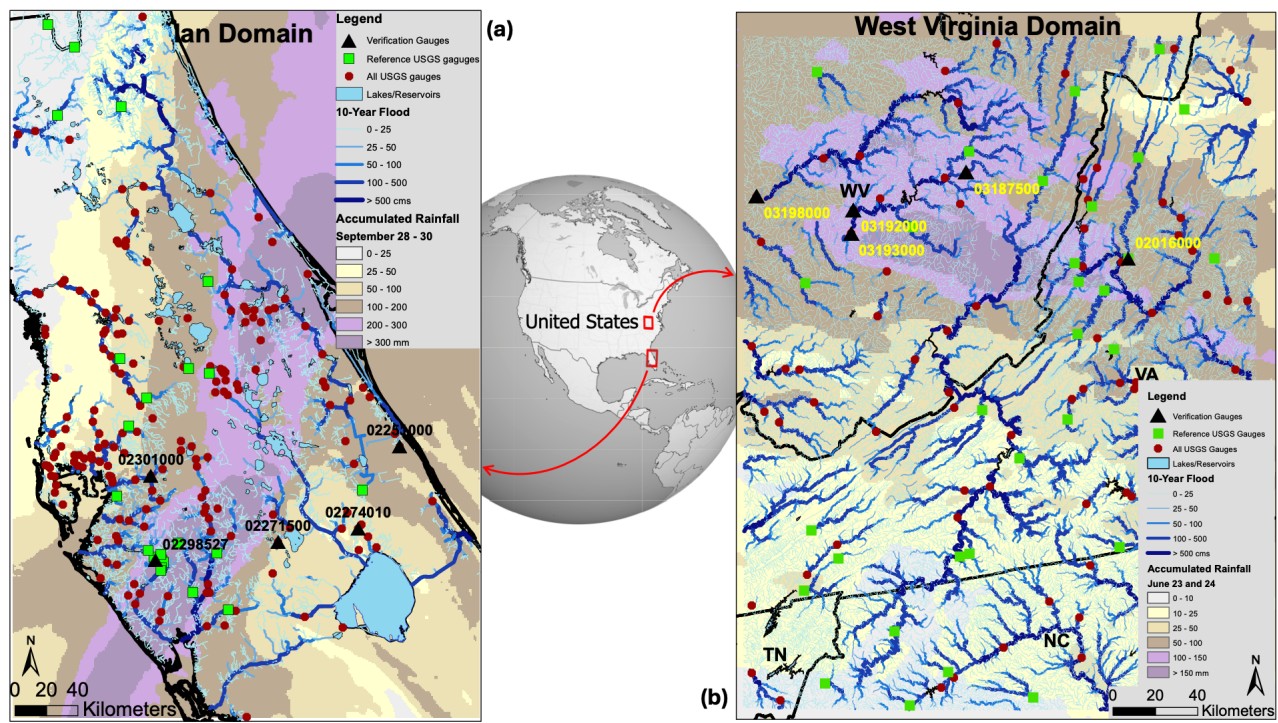

**Figure 2.** Study domains for (a) Florida's flood due to hurricane Ian and (b) West Virginia's flood. The background map for Ian depicts the accumulated rainfall drawn from NWMv2.1 Analysis and Assimilation atmospheric forcing during September 28 through 30, 2022. In (b), the colored map depicts accumulated rainfall drawn from Analysis of Record for Calibration (AORC) atmospheric forcing during June 23 and 24, 2016. Stream reaches in both domains are color coded based on their 10-year flood magnitudes calculated from the 42-year NWMv2.1 retrospective model simulation. USGS streamflow observation gauges are shown in red circles. GAGES-II reference streamflow gauges are shown by green rectangles, and five verification gauges are denoted by black triangles each with their 15-digit gauge identifiers. Lakes and reservoir bodies are shown as blue polygons.

## 3.2 Observations

Hourly or sub-hourly streamflow observations, with frequencies as high as 5 minutes, are gathered from the USGS streamflow measurement sites. Our data collection involves accessing observations well after the events, and we do not rely on near-real-time data acquisition. Consequently, discharge estimates have been updated through subsequent quality controls, including potential revisions to rating curves to better align with extreme flooding events and out-of-bank flows. For our West Virginia and Florida's Hurricane Ian test cases, streamflow observations were downloaded on 01/11/2022 and 01/30/2023, respectively. In the case of FL's Hurricane Ian, some gauges still lack event information in the observations, and the data remains provisional even months after the occurrence. The assimilation is conducted hourly in both domains, with sub-hourly USGS data averaged before the assimilation process begins.

To model streamflow observation errors, we adopt a heteroscedastic Gaussian error model with zero mean and a standard deviation set to 25% of the observed flow, following Abbaszadeh et al. (2019). Given that the gauges are situated on the stream network, this results in a linear observation operator $\mathbf{H}$ which simplifies the Kalman update, excluding nonlinear issues from the analysis step of HydroDART. Additionally, we assume that observations and their associated errors are uncorrelated in both space and time, yielding a diagonal observation error covariance matrix, $\mathbf{R}$.

Within the HydroDART framework, observations falling beyond 3 total standard deviations (equivalent to the square root of the sum of the prior ensemble and observation error variance) from the prior ensemble mean are rejected. This threshold, determined by the outlier threshold parameter in DART, serves multiple purposes. It helps avoid assimilating inaccurate observations, and it prevents the inclusion of observations where the mean of the ensemble members is quite far from the observation value. Such assimilation could lead to unacceptably large increments, potentially destabilizing the model run.

### 3.3 Retrospective Model Simulations

The NOAA National Water Model version 2.1 offers a publicly accessible multi-decade retrospective simulation covering the Contiguous United States (CONUS). This simulation is based on the NWMv2.1 model code and static files, driven by AORC atmospheric forcings. Notably, the retrospective simulation lacks data assimilation within the hydrologic model. While serving as a valuable resource for historical modeling context and real-time operations, it also facilitates the assessment of flow frequencies and model verification over an extended period. The 42-year retrospective simulation for NWMv2.1, spanning from February 1979 to December 2020, is openly available in two formats: Network Common Data Form (NetCDF) and Zarr on Amazon Web Services (AWS). Interested users can access the data at the following link: https://registry.opendata.aws/nwm-archive/.

In our approach, we leverage the 42-year NWMv2.1 retrospective simulations to construct the static background covariance matrix, denoted as $\mathbf{B}$, for our hybrid ensemble-variational filter. To achieve this, we assemble a 1,000-member ensemble from the retrospective model simulation. While the climatological ensemble could potentially be larger given the extensive data volume in the retrospective simulation, we opt for 1,000 samples to manage storage and computational costs. Previous research has explored the climatological ensemble size in the context of hybrid ensemble-variational data assimilation, indicating that an ensemble on the order of hundreds to a few thousands is generally sufficient to match the mean correlation length scales of $\mathbf{B}$ (e.g., Lei et al., 2021). It's important to note that the climatological ensemble is not advanced forward in time.

The climatological ensemble is subjected to an offline Empirical Orthogonal Function (EOF) analysis. The purpose of this analysis is to determine the spatial patterns and correlations present in the ensemble. This exploration aids in understanding the variability captured by the climatology and its potential impact on the hybrid ensemble-variational filter. The EOF analysis reveals that employing 1,000 static members results in a covariance matrix free from noise, where large-scale patterns dominate the initial EOFs, and smaller-scale patterns are encapsulated in the latter EOFs. This information is instrumental in shaping our understanding of the climatological ensemble's composition and behavior.

In terms of implementation, for Hurricane Ian, the process involves extracting two state variables, streamflow and water depth, from the dataset encompassing all 42 years for all reaches and buckets in the domain. A temporal subset is then created,

335 focusing exclusively on the month of September, given Ian's occurrence during that period. Every model simulation within the month of September from these 42-year model simulations is considered a plausible member of the climatology ensemble. Because the model simulation is at an hourly temporal scale, there are 42 (number of years) × 30 (number of days in September) × 24 (number of hours in a day) = 30,240 realizations to choose from. Subsequently, 1,000 members are randomly selected from this dataset and preserved for use by HydroDART, as outlined in equation (7). For West Virginia, a parallel approach is

340 adopted, with the exception that the model simulations from the month of June are utilized in constructing the static/climatology members. It's important to note that, for the purpose of constructing the climatology, we exclude the year 2016 in West Virginia to ensure that the flooding scenario under observation is not part of the climatology. In the case of Hurricane Ian, the year 2022 is not part of the retrospective run, eliminating the need for its exclusion.

### 3.4 Experimental Design and Verification

To test the performance of the hybrid scheme against the EnKF, we perform different assimilation runs in which we set the ATS localization cutoff distance to 100 km and turn on adaptive prior and posterior inflation following El Gharamti et al. (2021).

Our experiments commence in Section 4.1 by testing the performance of the EnKF within the HydroDART framework using an ensemble of 80 members. This approach is similar to the experiments outlined in the prior HydroDART study which focused on hurricane Florence in North Carolina (El Gharamti et al., 2021). The objective is twofold: to assess the prediction system's

performance in distinct basins characterized by diverse modeling and precipitation complexities, and to establish a baseline performance, both qualitatively and quantitatively, for the EnKF. This baseline serves as a reference point from which we intend to enhance predictive capabilities through the implementation of the hybrid approach in subsequent sections. For the hybrid EnKF-OI runs, we first examine the sensitivity of the scheme with respect to a few constant choices of the weighting factor (Section 4.2) and explore the impact of hybridizing the background covariance on the ensemble spread and inflation (Section

4.2.1). The idea is to determine whether the inclusion of climatological covariances would nullify the use of inflation in the dynamic ensemble. After determining the optimal hybrid weight (Section 4.2.2), several sensitivity runs with respect to the size of the dynamic ensemble are conducted (Section 4.3). Those runs are aimed to uncover the computational characteristics of the hybrid scheme and figure out whether it can be run more efficiently than the EnKF. The adaptive variant of the hybrid EnKF-OI algorithm is then studied in detail in Section 4.4. Finally, the application of the adaptive scheme for short-range forecasts is

investigated in both domains in Section 4.5.

To assess the quality of the estimated streamflow, we use many ensemble and hydrological metrics as shown in Table 1. Some of these metrics are deterministic in nature such as the root mean square error (RMSE) and others are probabilistic such as the reliability index (RI). We provide summary statistics by aggregating a few of these metrics for all flow gauges using boxplots. Where necessary, two sample t-tests are conducted to comment on the statistical significance of one experimental

result over others. The metrics are also computed separately for individual hydrographs, low flows and high flows. Low flows are characterized by computing the $10^{th}$ percentile of observed streamflow at each gauge over the entire assimilation period, while anything above the $90^{th}$ percentile is deemed a high flow. From Table 1, the centered root mean square error is used to construct Taylor diagrams (Taylor, 2001) which offer a comprehensive view for all gauges in the present hydrologic domains.

For optimal performance, Taylor diagrams are generally characterized by a correlation of 1, with both C-RMSE and standard
deviation equal to 0. Rank Histograms (Anderson, 1996) are also utilized to study the reliability of the predicted streamflow
ensembles. Flat rank histograms often indicate reliable predictions while skewed ones usually hint to limited ensemble spread
and poor coverage of the observation.

## 4   Results

### 4.1   EnKF Runs

Figure 3 illustrates the estimated streamflow for Sebastian River in FL and Kanawha River in WV. The hydrographs depict the
evolution of observed and OL discharge, overlaid with the prior and posterior estimates. The South Prong St. Sebastian River,
located on the eastern side of FL, witnessed a rapid surge in streamflow on September 28, 2022, coinciding with the landfall
of Hurricane Ian. This flooding persisted for about a week before returning to normal flow conditions. In comparison to the
OL, both the prior and posterior ensemble estimates exhibit improved accuracy, offering a better representation of observed
flows, particularly during the flooding event. On average, the prior and posterior streamflow estimates are 45% and 51% more
accurate, respectively (measured in terms of RMSE), compared to the OL. Additionally, the NSE and KGE values, derived
from the EnKF ensemble mean, are notably higher and closer to 1 than those associated with the OL. This underscores the
substantial improvement in prediction achieved through the assimilation of streamflow data with the EnKF. Adaptive inflation
plays a crucial role in this success, dynamically growing during the flooding event to enhance ensemble spread and refine its
conformity to observed data.

Kanawha River in WV underwent severe flooding in our case study, with discharge values soaring to around 4000 cms.
Similar to FL's Ian case, the EnKF demonstrated superior alignment with streamflow observations compared to the OL. No-
tably, the EnKF estimates displayed a precise correspondence with the falling limb of the hydrograph. While excelling on the
rising limb, the EnKF fell short of assimilating all data due to a pronounced underestimation of streamflow. In this instance,
the EnKF's prior ensemble yielded an approximately 31% lower RMSE than the OL.

Figure 4 presents comprehensive summary diagnostics for both the Ian and WV flooding cases, utilizing data from both 'all'
and 'reference' USGS gauges within the domains. The 'reference' designation pertains to the GAGES-II reference gauges.
Estimates derived from the OL and the EnKF at the reference gauges exhibit enhanced accuracy compared to those from all
gauges. This improvement is attributed to the fact that reference gauges are subject to little to no regulation and constitute only
a small subset of all gauges, resulting in fewer extreme outliers.

The CRPSS boxplots highlight the added value of the DA system over the OL, evident in positive scores. The majority
of gauges achieve scores surpassing 0.5. The NSE and KGE scores from the EnKF notably outperform those of the OL. It's
important to emphasize that the KGE and NSE are not directly comparable metrics, despite having the same range of variation
(Knoben et al., 2019). For the OL run, numerous gauges yielded efficiency estimates less than -10 (omitted for figure clarity).
The summary boxplots for the RMSE demonstrate the superior assimilation performance of the EnKF in both flooding cases.
When averaging over all gauges and across time, EnKF predictions exhibit at least 50% and 30% greater skill than the OL for

**Table 1.** Performance metrics used for evaluating the streamflow ensemble estimates. $Q$ refers to discharge or streamflow, measured in cubic meters per second (cms). The superscript $f|a$ means either prior or posterior discharge and $o$ denotes the observed discharge. $N_t$ is the total time steps or DA cycles. The $\bar{}$ and $\widehat{}$ notations refer to averaging over the ensemble and time, respectively. OL is a term used to describe the open loop or the unconstrained ensemble model run (without assimilation). $F$ and $H$ denote the cumulative distribution function (CDF) and the Heaviside function, respectively. CRPS is the continuous ranked probability score (Matheson and Winkler, 1976).

| Name | Equation | Description |
|------|----------|-------------|
| Root Mean Squared Error (RMSE) | $\dfrac{1}{N_t}\sum\limits_{k=1}^{N_t}\sqrt{\dfrac{1}{N_e}\sum\limits_{i=1}^{N_e}\left(Q_k^{f|a,i}-Q_k^o\right)^2}$ | Deterministic metric that varies between 0 and $\infty$, with a perfect score of 0. It measures the average distance between the predicted ensemble members and the observed discharge. |
| Ensemble Spread (ES) | $\dfrac{1}{N_t}\sum\limits_{k=1}^{N_t}\sqrt{\dfrac{1}{N_e}\sum\limits_{i=1}^{N_e}\left(Q_k^{f|a,i}-\overline{Q}_k^{f|a}\right)^2}$ | A deterministic measure of the variability of the ensemble, varying between 0 and $\infty$. |
| Mean Absolute Error (MAE) | $\dfrac{1}{N_t N_e}\sum\limits_{k=1}^{N_t}\sum\limits_{i=1}^{N_e}\left|Q_k^{f|a,i}-Q_k^o\right|$ | A deterministic measure similar to the RMSE but better suited for the case of non-Gaussian errors, using the L1-norm. |
| Continuous Ranked Probability Skill Score (CRPSS) | $\mathrm{CRPSS}=1-\mathrm{CRPS}^f\left(\mathrm{CRPS}^{\mathrm{OL}}\right)^{-1},$ <br><br> $\mathrm{CRPS}=\int\limits_{-\infty}^{+\infty}\left(F(Q)-H\left(Q\geq Q^o\right)\right)^2 dQ$ | A probabilistic metric $\in(-\infty,1]$ that computes the added skill by DA over the OL (Hersbach, 2000). A CRPSS of zero means that DA didn't improve the prediction skill of the model. CRPSS = 1 is a perfect score. Negative CRPSS values indicate that DA yields worse predictions than the OL. |
| Nash-Sutcliffe Efficiency (NSE) | $1-\dfrac{\sum_{k=1}^{N_t}\left(\overline{Q}_k^f-Q_k^o\right)^2}{\sum_{k=1}^{N_t}\left(Q_k^o-\widehat{Q}^o\right)^2}$ | A deterministic metric, that varies $(-\infty,1]$, to quantitatively assess the similarity between the estimated and the observed flow (Nash and Sutcliffe, 1970). The closer NSE to 1, the more accurate the estimated flow is. |
| Kling-Gupta Efficiency (KGE) | $\mathrm{KGE}=1-\sqrt{(r-1)^2+(\xi-1)^2+(\delta-1)^2}$ | A deterministic measure combining correlation coefficient $r$, bias $\xi$ and flow variability $\delta$ (Gupta et al., 2009). It varies between $(-\infty,1]$ and it provides a statistically more concrete metric than NSE. |
| Coefficient of Variation (CV) | $\left(\sum\limits_{k=1}^{N_t}Q_k^o\right)^{-1}\sqrt{N_t\sum\limits_{k=1}^{N_t}\left(Q_k^o-\widehat{Q}^o\right)^2}$ | A deterministic metric provides a concise measure of the variability in the observed flow. CV = 0 refers to a constant flow. |
| Centered Root Mean Squared Error (C-RMSE) | $\sqrt{\dfrac{1}{N_t}\sum\limits_{k=1}^{N_t}\left[\left(\overline{Q}_k^f-\dfrac{1}{N_t}\sum\limits_{k=1}^{N_t}\overline{Q}_k^f\right)-\left(Q_k^o-\widehat{Q}^o\right)\right]^2}$ | Deterministic measure used to aggregate estimates from different gauges in a single metric. |
| Reliability Index (RI) | $1-2\left[\dfrac{1}{N_t}\sum\limits_{k=1}^{N_t}\left|F_k\left(Q_k^\circ\right)-U\left(Q_k^\circ\right)\right|\right]$ | Probabilistic metric (Renard et al., 2010) varying between 0 and 1, and is used to quantify how close the empirical CDF of the observed flow to the uniform distribution, $U$. RI = 0 is the worst and RI = 1 reflects perfect reliability. |

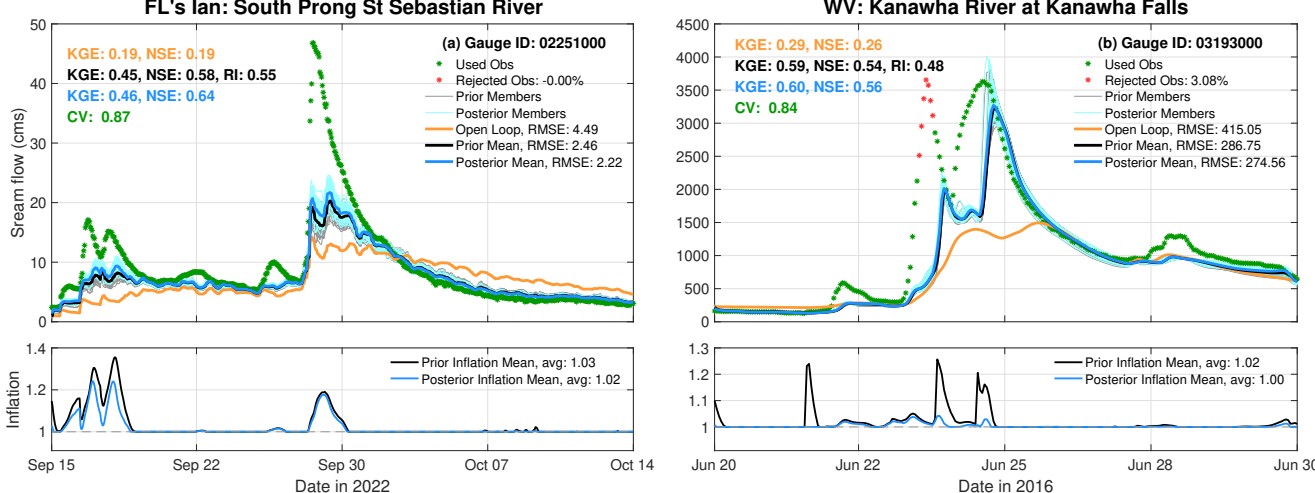

**Figure 3.** Left: Streamflow time-series (hydrograph) during hurricane Ian at South Prong Street Sebastian River. Observed streamflow is given by hourly asterisks. Green asterisks indicate that the observation was assimilated. The rejected observations due to outlier threshold are shown in red asterisks. OL streamflow estimates are given by the orange curve. Prior and Posterior ensemble means are denoted by the thick black and blue curves, respectively. The light gray and cyan curves show the prior and posterior ensemble members, respectively. Time-averaged streamflow RMSE values are reported in the legend. Other metrics such as CV, NSE, KGE and RI are also annotated. The evolution of prior and posterior inflation over time is shown in the bottom panel. The right panels are similar to the left ones but for the Kanawha River at Kanawha Falls in WV.

the Ian and WV flooding cases, respectively. This was also supported through a statistical significance test using a two-sample t-test. The test (not shown) rejects the null hypothesis which states that the RMSE averages resulting from the open loop and the HydroDART priors are equal. In the subsequent sections, we will delve into the hybrid results to explore how they build

upon the achievements of the EnKF presented in this section.

## 4.2   Hybrid Runs

We examine five cases where we fix the hybrid weighting coefficient both in space and time, setting $\alpha$ to values of 0.1, 0.3, 0.5, 0.7, and 0.9. It's crucial to note that, from eq. (7), $\alpha$ represents the weight on the dynamic background error covariances, while $(1-\alpha)$ signifies the weight on the static background. Therefore, the chosen values of $\alpha$ progressively emphasize the dynamic

background more in order. Experiments involving adaptive hybrid weighting are discussed further in section 4.4.

    Figure 5 displays hydrographs for the North Prong Alafia River at Keysville, FL. Situated just outside Tampa on the western corridor of the state, this gauge provides valuable insights. All EnKF-OI runs exhibit an improved fit to the hydrograph peak on September 29. Both the standard EnKF and the OL underestimate the flooding event, predicting a very delayed recession. Beyond the primary flooding event, the hybrid runs precisely capture the rising and falling limbs of the hydrographs. The

EnKF-OI with $\alpha = 0.5$ stands out as the most accurate, yielding a KGE value of 0.92. However, as $\alpha$ approaches 1, the

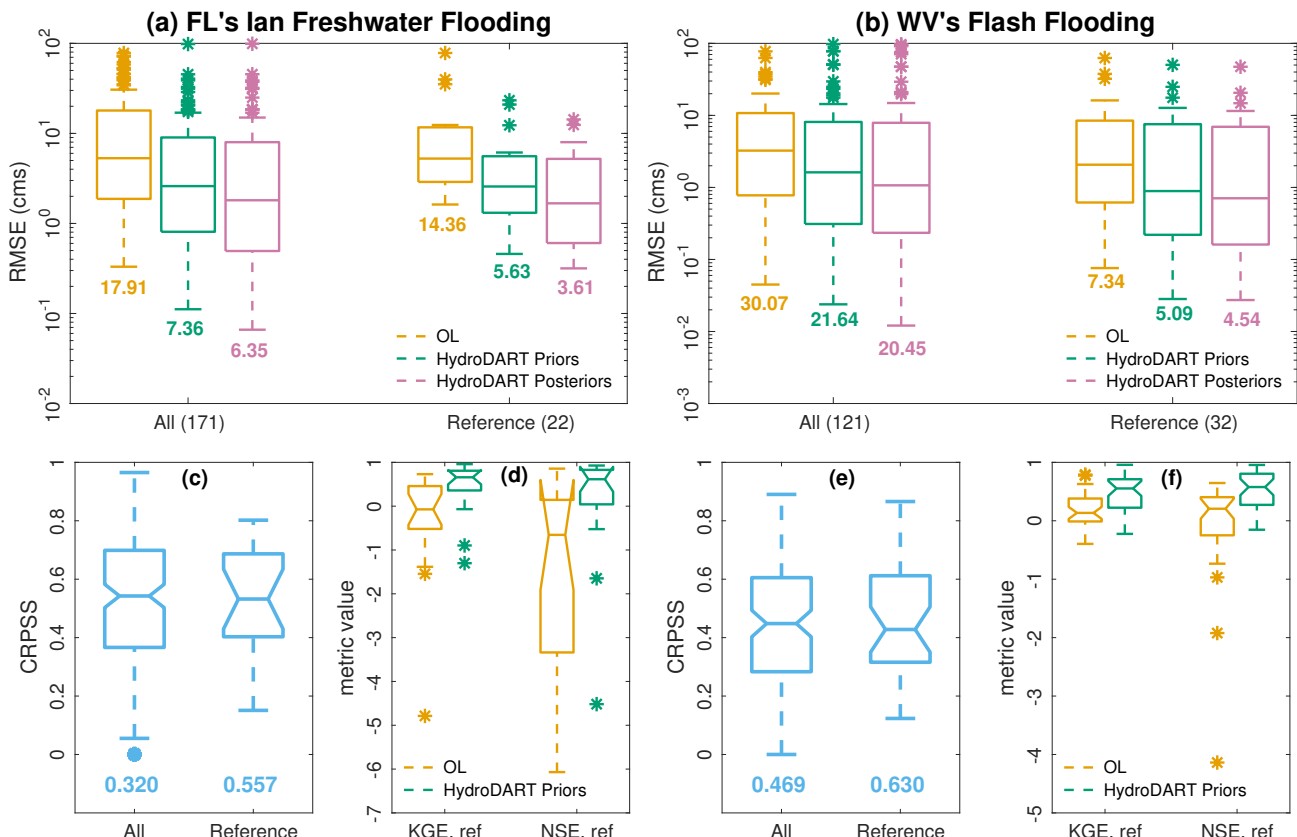

**Figure 4.** Top left: RMSE boxplots for OL and EnKF prior and posterior estimates for all gauges and for GAGES-II reference USGS gauges in the FL domain. The number of all and reference gauges in both domains are given in the labels of panels (a) and (b). Note that the y-axis is in log-scale. Averaged RMSE are reported underneath the individual boxplots. Bottom Left: CRPSS boxplots (blue) for all gauges and for GAGES-II reference gauges. NSE (green) and KGE (yellow) boxplots are also shown for the reference gauges. The right panels are similar to the left ones but for the WV flash flood case.

performance starts to degrade due to the heavier weight placed on the dynamic sample covariance. Conversely, when $\alpha$ is close to 0, time-dependent information in the background covariance is limited. For instance, in the case of $\alpha = 0.1$, both the prior and posterior ensemble spread become quite large. This unusual increase in ensemble spread is a result of the dominance of the climatological ensemble over the dynamic one.

Figure 6 provides a comparison between the hybrid EnKF-OI scheme and the standard EnKF at the Kanawha River gauge near Charleston, WV. This area was affected severely by the flooding event under consideration. On June 23rd, the river's discharge rose to almost 5000 cms. The OL, representing the hydrologic model, fails to capture the flooding event entirely. While the EnKF struggles to accurately predict the initial day, it improves toward the 24th by simulating the end of the flood peak and its recession. However, given the intensity of the flood peak, the EnKF estimates, while surpassing the OL, are not

satisfactory (NSE $\approx$ 0). In contrast, the hybrid EnKF-OI solution, particularly with $\alpha \leq 0.5$, more accurately simulates the

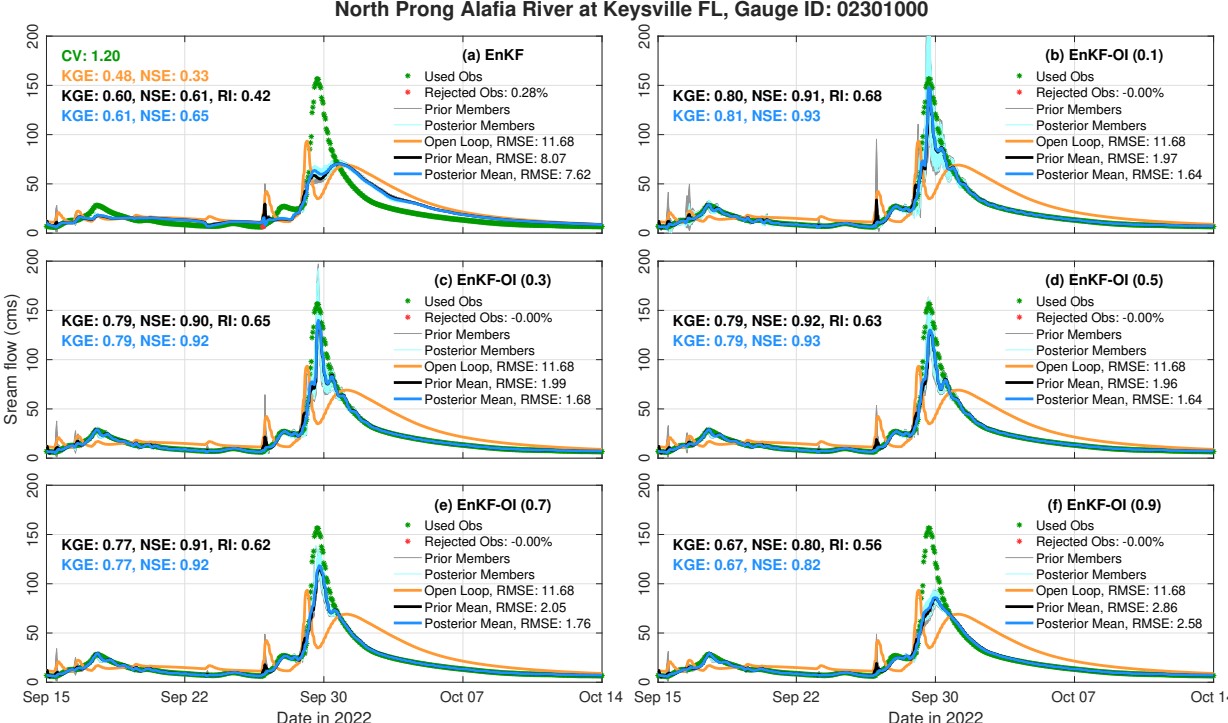

**Figure 5.** Hydrographs for North Prong Alafia River in FL using the EnKF and the 5 hybrid EnKF-OI runs. The hybrid runs use different weighting coefficients $\alpha$ reported in the title of the individual legends. Similar to Fig. 3, CV, NSE, KGE, RI, and RMSE values are reported on all panels.

observed streamflow, resulting in an NSE exceeding 0.95. Increasing the hybrid weight beyond 0.5 diminishes the EnKF-OI's skill, although it still shows superior performance compared to the EnKF (bottom panels). Remarkably, such performance can be achieved by incorporating only 10% (i.e., $\alpha = 0.9$) of the hybrid covariance from the climatology in panel (f). Since the climatological covariance is not susceptible to sampling errors, its partial integration into the EnKF helps mitigate significant

model biases, especially during flooding events. This is also apparent from the RI measures which tend to shrink as the impact of **B** decreases.

### 4.2.1   Ensemble Spread and Inflation

The hybrid covariance approach provides the dynamic ensemble of the EnKF with more diverse correction directions allowing for a better analysis. A crucial aspect of this hybridization is the augmentation of ensemble variability, a factor anticipated

to improve performance in the presence of model uncertainties and forcing biases. To illustrate this, Figure 7-(a) depicts the Average Ensemble Spread (AES) across all gauges for both study domains. Given the unconstrained nature of the OL, it is reasonable for its AES to be larger than that of the EnKF. The EnKF, due to its hourly analyses, experiences substantial shrinkage of the ensemble spread. This becomes problematic when the shrinkage occurs far from the observations, leading

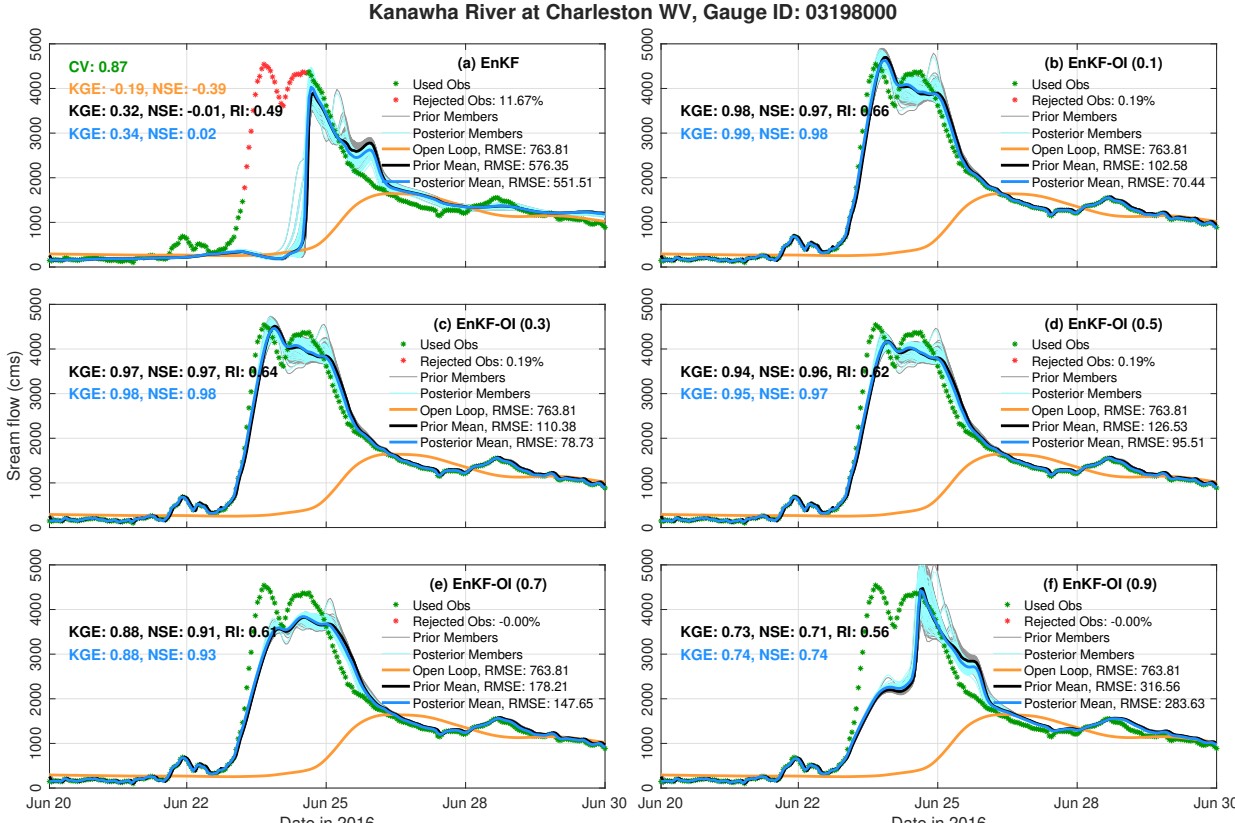

**Figure 6.** Similar to Fig. 5 but for Kanawha River at Charleston in WV.

to poor streamflow estimates, especially in the presence of model biases. While inflation is often employed to address this
issue, it may not always effectively counter biases. For $\alpha = 0.1$, most of the weight is placed on the climatological information
resulting in a notable increase in the ensemble spread. This is because the climatological covariance **B** comprises a substantial
inventory (1,000 instances over 42 years) of historical streamflow distributions, contributing to its relatively large variance. As
$\alpha$ increases, the ensemble spread decreases until it aligns with the EnKF spread for $\alpha = 0.9$.

   The reliability of the predicted streamflow estimates is analyzed in panels (b) and (c) of Figure 7 using rank histograms. At
Ogleby Creek in FL, the discharge ensemble obtained using the EnKF is slightly overdispersive. The hybrid scheme with 0.3
weight, on the other hand, displays a flat rank histogram indicating better reliability. This also means the flow members resulting
from the EnKF-OI scheme are indistinguishable from the observed flow. At Cowpasture River in the second domain (panel c),
a large fraction of the observations (nearly 40%) appears to be larger than the simulated discharge indicating underestimation.
The rank histogram of the EnKF also shows partial skewness to the right. The hybrid scheme successfully mitigates that bias
and yields an improved ensemble spread. Assessment of the reliability at other locations offered a similar conclusion.

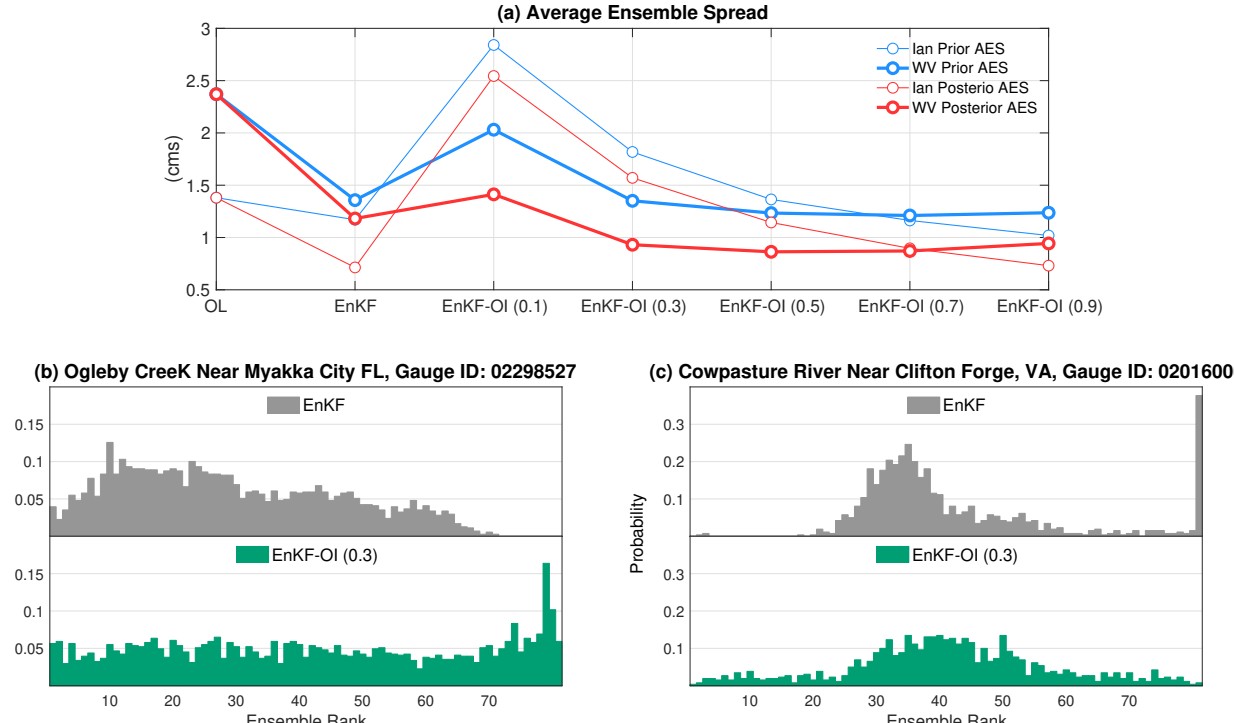

**Figure 7.** (a) Average Ensemble Spread, AES across all gauges for the OL, EnKF and the hybrid EnKF-OI with 5 distinct weights. Results are shown for FL (thin) and WV (thick) for the prior and the posterior ensembles. (b) and (c) show rank histograms for streamflow at different locations in both test domains. The rank histograms are displayed for the EnKF and the hybrid EnKF-OI with weighting set to 0.3.

Figure 8 illustrates the time evolution of prior inflation for the same gauges examined in Figs. 5 and 6. At North Prong Alafia River in FL, the hybrid EnKF-OI runs utilize inflation solely during flooding events, turning it off completely before and after the rainfall events. Early inflation spikes around Sep. 16 and 17 may address modeling bias and limited initial variability. For Kanawha River in WV, inflation is predominantly unused by the hybrid runs, except for a brief initial period on June 20$^{\text{th}}$. In contrast, the EnKF employs inflation not only during flooding periods but also in non-flooding intervals. Averaging across all gauges, the hybrid EnKF-OI with $\alpha = 0.5$ requires approximately 5 to 10% less inflation than the EnKF, a trend consistent with posterior inflation. Consequently, we opted to retain the inflation algorithm for the hybrid runs. In summary, we posit that inflation primarily addresses instantaneous bias, while the inclusion of static background information serves to alleviate long-term biases and uncertainties.

### 4.2.2 Optimal Hybrid Weight

The Taylor diagrams and RMSE boxplots in Figure 9 offer a comprehensive view for all gauges within the FL and WV domains. Given the multitude of gauges in each domain, the Taylor diagram exhibits a cloud of points, one for each streamflow gauge. Visually, the cloud of points corresponding to the hybrid runs tends to cluster closer to the ideal performance point than

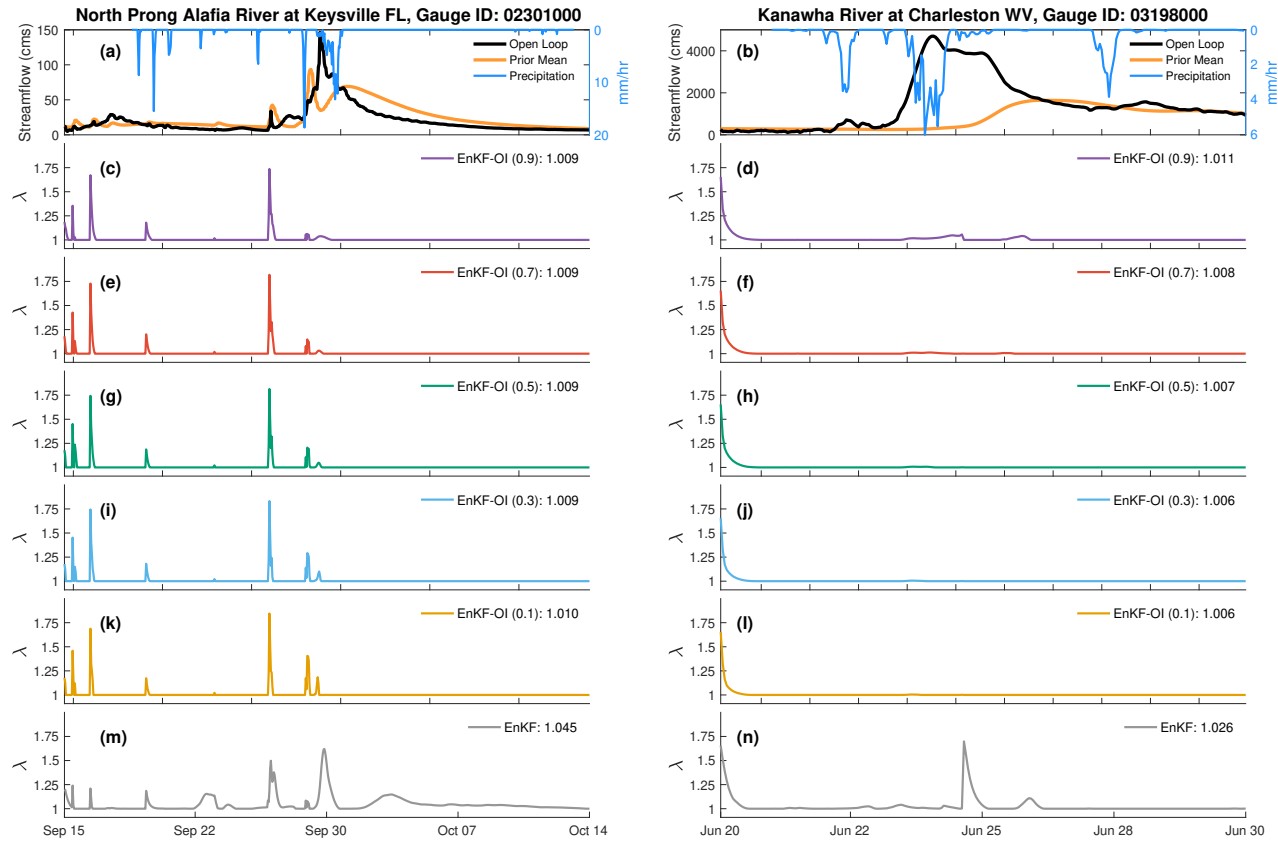

**Figure 8.** Time-series of prior inflation resulting from the EnKF (m, n) and the hybrid EnKF-OI with constant weights (panels c-i). The inflation, denoted by $\lambda$, in equation (4) is adaptive in space and time. North Prong Alafia River in FL is represented by the left panels. The right panels show the results for Kanawha River at Charleston in WV. Panels a, b show the open loop and EnKF-OI (0.1) streamflow in addition to mean areal precipitation in mm/hr.

the cloud for the EnKF. For instance, in the WV case, the EnKF cloud is perceptibly closer to the left side of the diagram,
indicating lower accuracy compared to other schemes. Aggregating results across all gauges, the resulting correlations for each scheme are tabulated in Table 2. In both domains, correlations from the hybrid runs surpass those of the EnKF. The EnKF-OI schemes yield comparable correlations, with $\alpha = 0.7$ and $0.5$ offering the best performance in FL and WV, respectively. The boxplots in Figure 9 echo a similar narrative-the hybrid EnKF-OI enhances prior RMSE by over 50% compared to the EnKF. Among the tested hybrid runs, $\alpha = 0.1$ contributes to the least accurate results, underscoring the significance of maintaining a
dynamic ensemble in the filtering framework for skillful streamflow estimation.

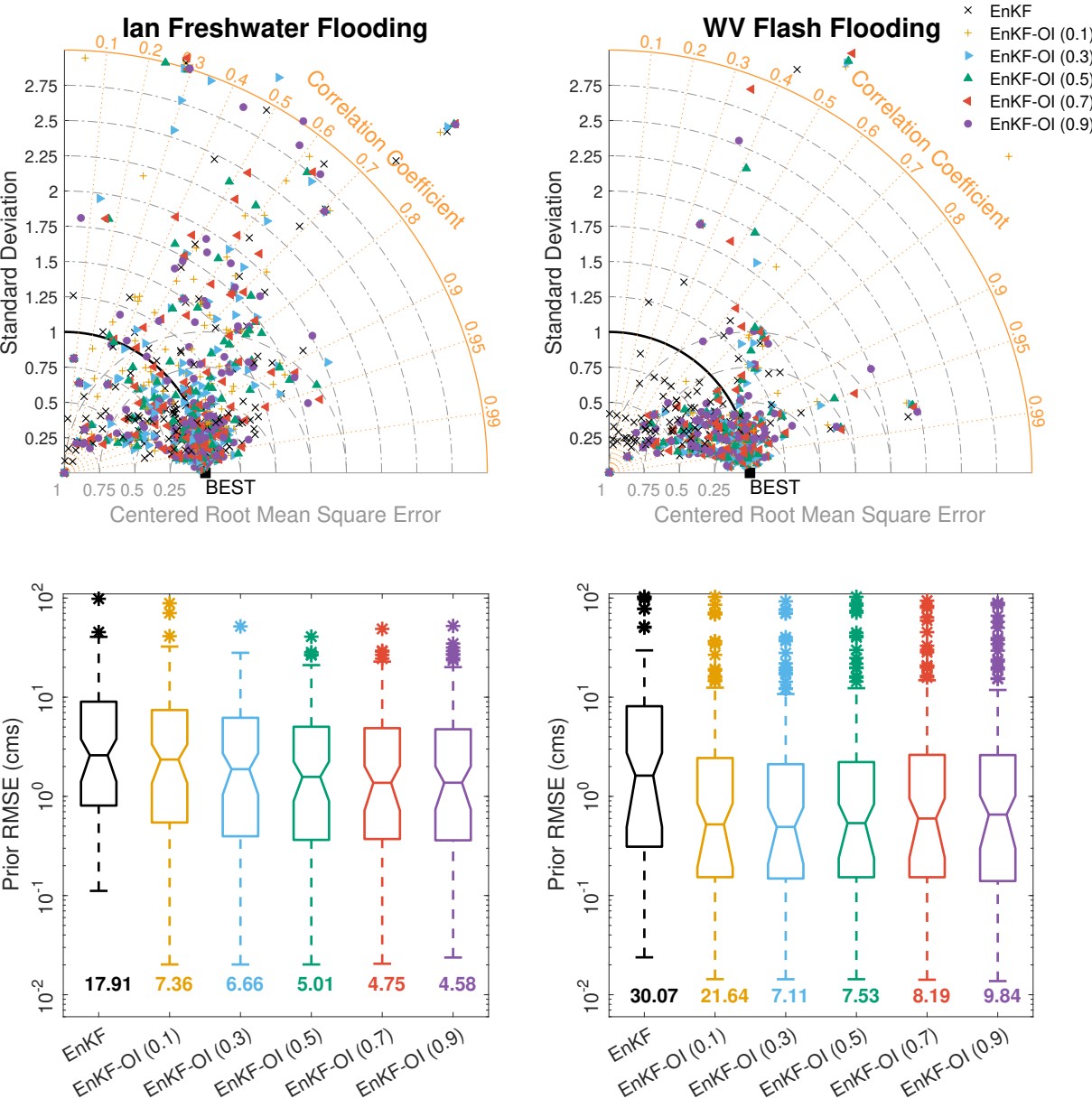

**Figure 9.** Top panels: Taylor diagrams for Ian's flood (left) and WV's flash flood (right). Each cloud of points on the diagram denotes all gauges available in the domain. "BEST" is the point of ideal performance, having perfect correlation with the observed flow and zero error and standard deviation. Gauges with standard deviations larger than 3 are omitted for visual purposes. Bottom panels: Prior RMSE boxplots for all gauges in each domain, obtained using the EnKF and the hybrid scheme with fixed weights. Overall averaged prior RMSE in each case is reported beneath the individual boxplots.

**Table 2.** Pearson correlation between the observed discharge and the prior estimates suggested by the EnKF and the hybrid EnKF-OI with constant weights. The correlation is computed for all gauges in each domain and averaged over time.

| Domain | EnKF | EnKF-OI (0.1) | EnKF-OI (0.3) | EnKF-OI (0.5) | EnKF-OI (0.7) | EnKF-OI (0.9) |
|--------|------|---------------|---------------|---------------|---------------|---------------|
| FL | 0.625 | 0.723 | 0.757 | 0.758 | 0.766 | 0.756 |
| WV | 0.666 | 0.859 | 0.858 | 0.861 | 0.852 | 0.850 |

### 4.3 Dynamic Ensemble Size

The findings in Section 4.2.2 prompt a critical question: How large of an ensemble is necessary to ensure precise and efficient predictions?

Figure 10 illustrates the distribution of the mean absolute error for all USGS gauges and reference gauges separately for five different sizes $N_e = (10, 20, 40, 60, 80)$. Additionally, we compare the results with our baseline 80-member EnKF runs (without hybrid) in both domains. The observed prior MAE for the WV flash flooding is larger than that for FL flooding, attributed to the shorter duration of the WV event, making it inherently flashier. In comparison to the EnKF, the hybrid runs consistently exhibit improved bias, evident for both all and reference gauges. For instance, using only 10 members results in an averaged prior bias of 4.79 cms and 12.41 cms for FL and WV cases, respectively. This represents an average reduction of 29% (FL) and 40% (WV) in prior bias compared to the 80-member EnKF runs. A two sample t-test (with 5% significance level) was conducted to confirm that the differences between the runs are statistically significant. As shown in Figure 10-(a), the value of the t-statistic for all hybrid runs falls outside the cutoff region (area between the two dashed lines). This indicates that the reported MAE average for each EnKF-OI run is not equal to that of the EnKF thereby rejecting the test's null hypothesis. This was consistent for all runs in the WV case (not shown for figure clarity).

Among the hybrid runs, a dynamic ensemble of 80 members achieves the best performance in both cases. However, the difference in prediction accuracy between the EnKF-OI with 80 and 20 members is minimal ($\leq 6\%$). Notably, the computational cost of running the EnKF-OI with 20 members is approximately four times smaller than the run with 80 members. While reducing the ensemble size seems to slightly compromise prediction skill in our case studies, the trade-off of sacrificing 6% in accuracy, particularly evident in the 20-member EnKF-OI, is considered favorable for improving computational efficiency in operational settings. This conclusion is consistent across various metrics, including CRPSS, NSE, KGE, and RMSE.

### 4.4 Adaptive Hybrid Scheme

Building upon the ensemble analysis discussed in the preceding section, we opted for an ensemble size of 20 in our adaptive hybrid runs. The initial distribution for the hybrid weight ($\alpha$) was strategically chosen as a Gaussian random variable centered at 0.5, with a standard deviation of 0.005. This decision stems from insights gained in section 4.2, where our analysis revealed that a weight of 0.5 produced the optimal representation of prior streamflow in both domains. Given the adaptive nature of the

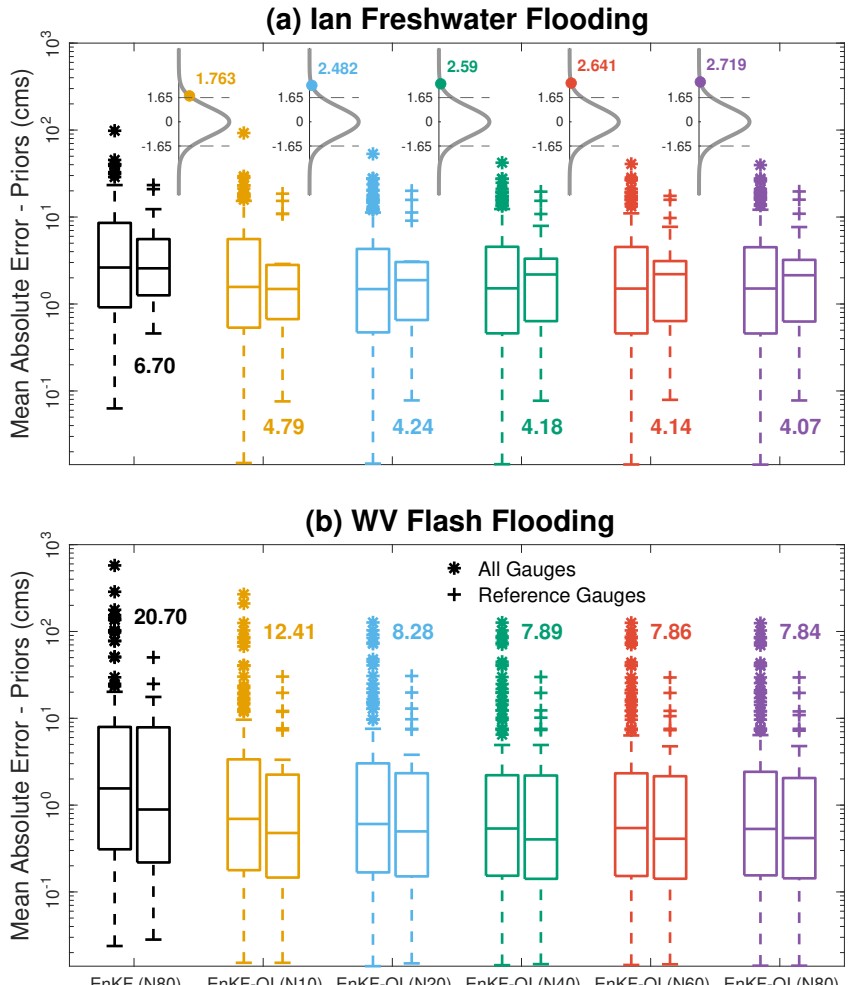

**Figure 10.** Prior mean absolute error (MAE) boxplots for all/reference gauges at Ian (a) and WV (b). The boxplots are obtained using the EnKF (with 80 members) and the hybrid scheme with different dynamic ensemble sizes. Time-averaged prior MAE for all gauges in each case is reported above the individual boxplots. Note that the y-axis is in log-scale. The pdfs (gray) of the two sample t-tests for each EnKF-OI run with respect to the EnKF are shown on top of the boxplots in panel (a). The t-statistics are denoted (and also annotated) by the circles and the dashed lines denote the critical cutoff.

algorithm, initiating DA with an equal weighting of dynamic and static covariances at 0.5 was deemed an intuitive starting point. In terms of the variance, we conducted several sensitivity experiments to assess the impact of the initial standard deviation of $\alpha$ on the accuracy and performance of the hybrid filter. Our findings indicated that the standard deviation primarily influences the speed at which the weight gets updated. Consequently, a standard deviation value of 0.005 was selected, as it yielded the most favorable overall behavior in our experiments.

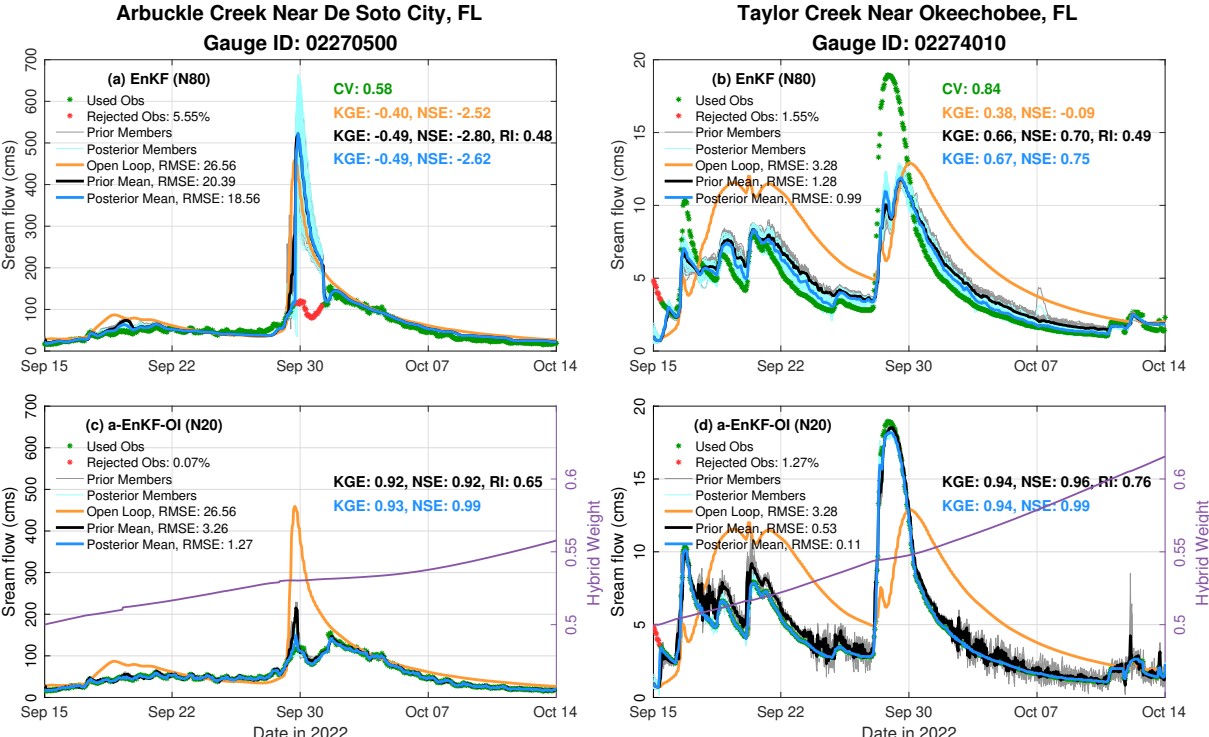

**Figure 11.** Hydrographs for Arbuckle Creek near De Soto City, FL (left) and Taylor Creek near Okeeechobee, FL (right). Top panels show the results from an 80-member EnKF together with the OL. The results from the a-EnKF-OI with 20 members are given in the bottom panels. For the a-EnKF-OI, the change in the hybrid weight over time is displayed in purple according to the right y-axis. Time-averaged metrics such as RMSE, NSE, KGE and RI are annotated on the individual panels.

Figure 11 displays hydrographs at Arbuckle and Taylor Creeks in FL, offering a comparative assessment of prediction performance between the EnKF with 80 members and the adaptive hybrid EnKF-OI (a-EnKF-OI) with a reduced ensemble size of 20 members. At Arbuckle Creek, the OL tends to overestimate the flooding event in late September by nearly 400 cms. While the EnKF excels before the main event, it mirrors the trajectory of the OL during the flood, rejecting 2 days

of data. In stark contrast, the a-EnKF-OI assimilates all observations, achieving a near-perfect fit to the observed discharge. Notably, the a-EnKF-OI attains high KGE and NSE values of 0.92, while the EnKF lags with scores of -0.49 and -2.80, respectively. The adaptive hybrid weight steadily increases from 0.5 to 0.55 during the assimilation period. Similarly, at Taylor Creek, the a-EnKF-OI performs quite well, yielding KGE and NSE values of 0.94 and 0.96, respectively. It further increases the reliability of the EnKF prior ensemble by 27%. Most improvements over the EnKF are evident during the last 2 days of

September, coinciding with the main flood peak. The lower discharge at Taylor Creek underscores the adaptive hybrid filter's consistency in varying hydrological conditions. The hybrid weight demonstrates an early increase, followed by a sharper rise on September $28^{th}$, reaching approximately 0.62 by the end of the simulation. The adaptive scheme's increased weighting on the dynamic ensemble, as reflected by the rising $\alpha$, signifies reduced bias and improved ensemble statistics. This shift

renders climatological information less crucial, indicating the hybrid scheme's adeptness at leveraging climatology to enhance ensemble bias and subsequently placing greater emphasis on the dynamic ensemble. In terms of computational efficiency, the 20-member a-EnKF-OI proves to be roughly 4 times more efficient than the 80-member EnKF, further highlighting the advantages of the adaptive hybrid filter.

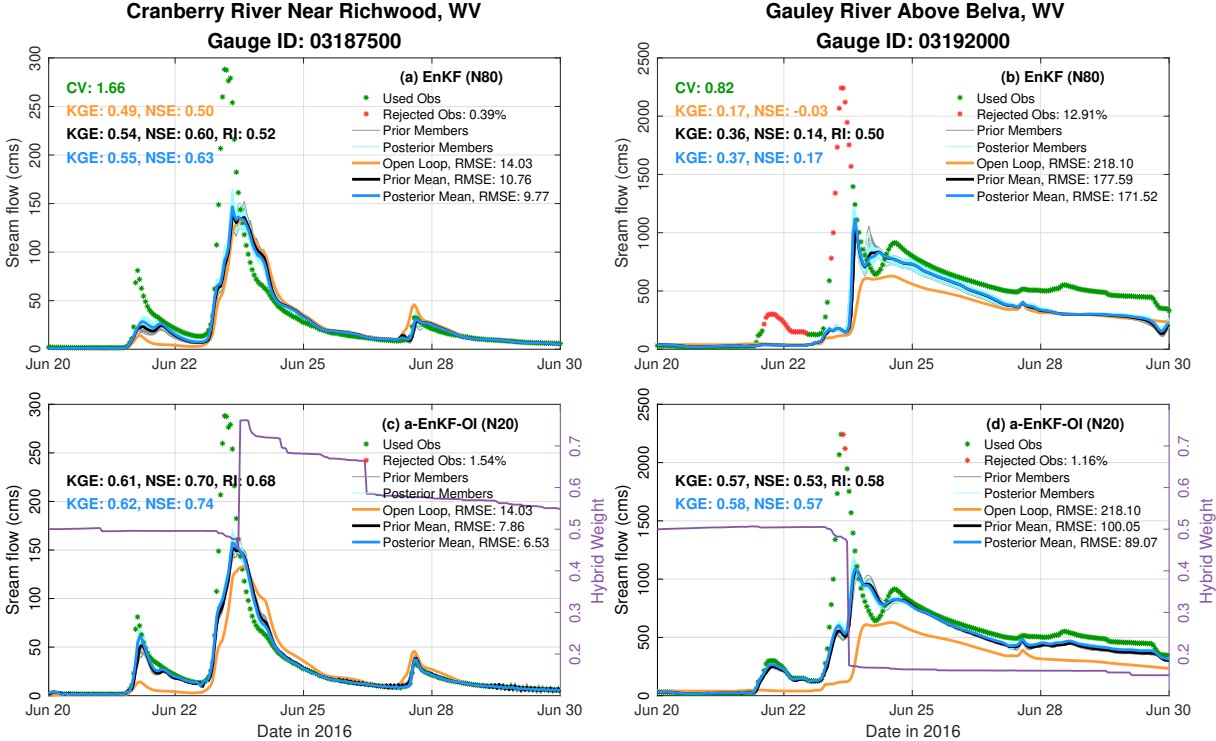

**Figure 12.** Similar to Fig. 11 but for Cranberry River near Richwood and Gauley River above Belva in WV.

The effectiveness of the 20-member a-EnKF-OI algorithm is further explored through its application to the flash flood events at Cranberry and Gauley Rivers in WV, as illustrated in Figure 12. At Cranberry River, both the EnKF and a-EnKF-OI schemes exhibit challenges in accurately capturing the observed discharge during the main flood peak. However, the a-EnKF-OI offers an improved prediction of the earlier event on June 21$^{st}$, 2016. Notably, the rising and falling limbs of the main event are more distinctly delineated using the hybrid filter. Both filtering algorithms demonstrate greater skill compared to the OL. During the recession period, the hybrid weight is elevated to 0.75 and subsequently decreases back to 0.55 on June 30$^{th}$. At Gauley River, the a-EnKF-OI notably outperforms the EnKF, particularly during the early hours of the flooding event and the subsequent recession period, resulting in an improved overall NSE of 0.53 compared to the EnKF's score of 0.14. The hybrid weight at this gauge undergoes a significant drop from 0.5 to almost 0.1 during the heavy rainfall event. This adjustment seems to be linked to the pronounced underestimation of streamflow. The adaptive scheme strategically places more weight on the climatology in an effort to alleviate the observed discrepancy between priors and observations. It's noteworthy that for both

gauges, the algorithm dynamically adjusts the hybrid weight, particularly around June 23rd, showcasing its responsiveness to
underlying prior ensemble biases.

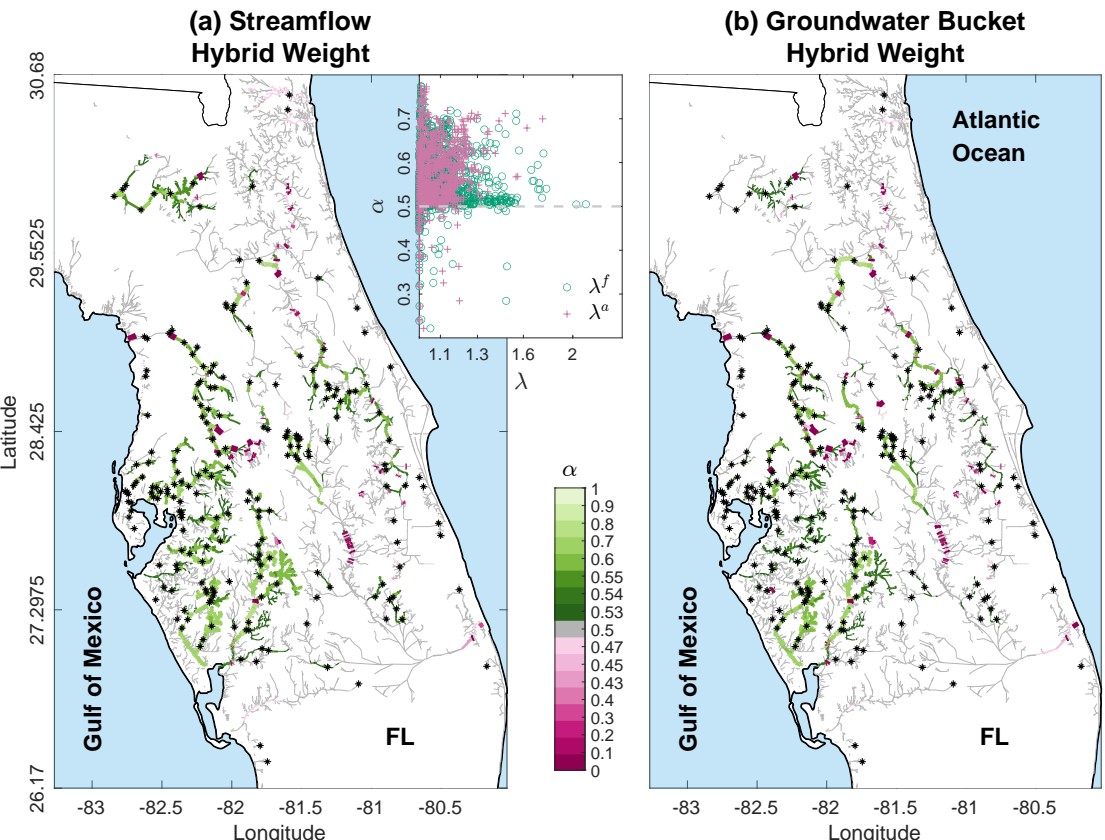

**Figure 13.** Spatial maps for streamflow (left) and groundwater bucket (right) hybrid weight $\alpha$. The maps are obtained on the 14th of October, 2022 at 11pm for Ian using the adaptive hybrid scheme. Black asterisks on the maps show the assimilated USGS gauges. The inset panel (top-middle) plots the time-averaged prior ($\lambda^f$) and posterior ($\lambda^a$) streamflow inflation on the x-axis and the streamflow hybrid weight on the y-axis (the same weight shown on the map). The initial starting point for the weight; i.e. $0.5$, is highlighted by the dashed gray line.

Figure 13 illustrates the spatial variations in hybrid weights for streamflow (panel a) and groundwater storage (panel b) on the last DA cycle, October 14, in the FL case. The most significant changes to the hybrid weights are concentrated in the proximity of observation points, aligning with the ATS localization strategy. In reaches far from observations, the hybrid weighting coefficients remain relatively stable, falling within the range of $0.48 < \alpha < 0.52$ (gray colored reaches). It's crucial to recall that ATS localization selectively influences reaches upstream and downstream from a given gauge, using a predetermined cutoff distance, in this case, set at 100 km. Streams with smaller hybrid weights generally indicate limited ensemble variability. A considerable number of streams exhibit increased $\alpha$, particularly on the western side of the state (green reaches) where Hurricane Ian made landfall. For example, Peace River near Fort Myers and its tributaries predominantly showcase weights

exceeding 0.6. The augmentation of $\alpha$ corresponds to an expectation that streamflow realizations derived from the hydrologic

model will align more closely with observed flow, as demonstrated in Figure 11. The adaptive hybrid scheme extends its spatial weight mapping to groundwater storage, an active participant in the assimilation process. The distribution of $\alpha$ for the buckets, as depicted in Figure 13-(b), mirrors that of streamflow. This suggests a non-zero correlation between discharge observations and groundwater storage, leading to multivariate updates in the hydrological state.

Examining streamflow, the most significant deviations from the initial weights (set at 0.5) are observed along streams where

inflation was minimal ($\lambda < 1.3$), as highlighted in the inset panel of Figure 13. This observation implies that $\alpha$ undergoes substantial updates in locations where inflation alone couldn't adequately address sampling errors and biases in the ensemble. The synergy between covariance hybridization and ensemble inflation techniques emerges as crucial for enhancing the quality of streamflow predictions. As demonstrated in section 4.2, while inflation tackles current streamflow conditions, long-term biases are effectively mitigated through the incorporation of climatological information – a facet where inflation alone falls

short.

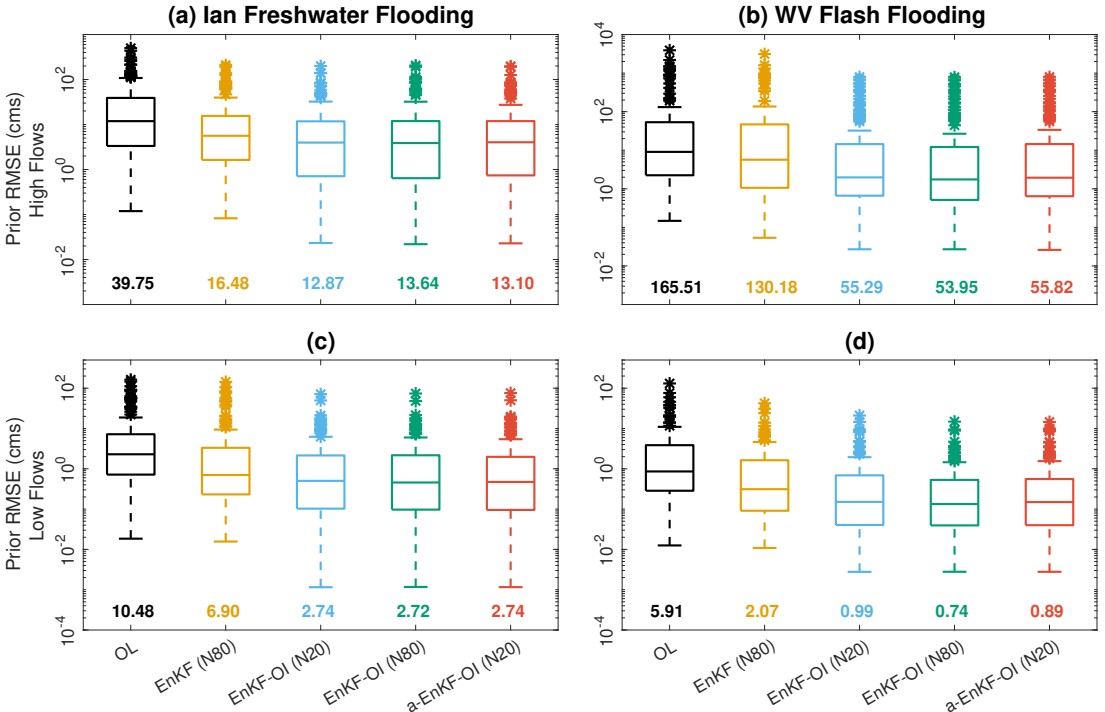

**Figure 14.** Boxplots for prior RMSE of upper decile (top panels) and lower decile (bottom panels) stream flows. 5 simulation runs are compared: OL, 80-member EnKF, 20-member EnKF-OI ($\alpha = 0.5$), 80-member EnKF-OI ($\alpha = 0.5$) and a 20-member a-EnKF-OI. Ian's flooding results are shown to the left while WV's flash flood estimates are shown to the right. Note that the y-axis is in log-scale. Averaged RMSE values are annotated underneath the individual boxplots.

The evaluation of the adaptive hybrid algorithm's performance is succinctly summarized for both low and high stream flows in Figure 14. The a-EnKF-OI with 20 members is compared against the OL, the original EnKF with 80 members, and the fixed weight EnKF-OI ($\alpha = 0.5$) with both 20 and 80 members. In FL, the most substantial improvements offered by the various hybrid schemes over the EnKF are observed for the lowest decile of flows. For instance, the average prior prediction of the a-EnKF-OI (2.74 cms) is 60% more accurate than the EnKF's score (6.90) when considering the lowest flow decile. Conversely, for the highest flow decile, the a-EnKF-OI (13.1 cms) exhibits a 21% gain in prediction skill over the EnKF (16.48), on average. In WV, the advantages of the adaptive hybrid scheme over the EnKF are comparable for both the highest and lowest decile flows. The intriguing behavior observed in the FL case can be attributed to two key factors. Firstly, the Ian simulation in FL spans a full month of streamflow analysis, wherein low-flow periods are more frequent than the main flooding event. Consequently, the performance differences between the schemes are expected to be more pronounced during low-flow periods, highlighting the OI-based scheme's benefits for both floods and non-peak flows. Secondly, unlike WV, where the model exhibits a strong positive bias compared to observed flow, in FL, the model generally suggests a negative bias. Compared to the 20-member EnKF-OI with a fixed $\alpha$, the adaptive variant demonstrates relatively similar accuracy with a slight advantage in WV's low-flow diagnostics. Overall, the fixed weight 80-member EnKF-OI consistently delivers the best RMSE scores for both low and high flows across the two hydrologic domains. The hybrid approach, encompassing various flavors, consistently yields substantial gains compared to the EnKF and considerable improvements compared to the OL. The 20-member a-EnKF-OI emerges as highly competitive, showcasing exceptional computational efficiency suitable for both flooding and non-flooding applications on larger domains.

## 4.5 Short-Range Reforecasts

The hybrid scheme and its adaptive variant have demonstrated substantial enhancements in streamflow simulations when contrasted with the OL and the EnKF. So far, our focus has been solely on verifying analyses (posteriors) and forecasts (priors) within the scope of hourly DA cycling. Put differently, we have yet to explore the influence of analyses beyond the one-hour forecast (prior) timescale. In this section, we delve into assessing the lasting impact of analyses on forecasts spanning up to 18 hours. We aim to unravel questions pertaining to the temporal persistence of state corrections in the model. Furthermore, we seek to identify forecast lead times at which the hybrid streamflow predictions exhibit enhancements over the OL.

To address these inquiries, we conducted a reforecast employing identical forcing data as utilized in the analysis cycles above. A comprehensive reforecast of the NWM would typically involve utilizing its forecast forcing datasets. It is acknowledged that these real-time forecasts might entail larger uncertainties compared to the analysis forcing datasets employed here. However, our primary focus lies in exploring the DA system's capacity to mitigate uncertainties in initial conditions and model biases. The decision to employ retrospective atmospheric forcing allows us to assess the impact of DA on enhancing initial conditions for forecast cycles, without being overshadowed by substantial uncertainties in the forcing dataset, as outlined by Rafieeinasab et al. (2014). Specifically, we employ the AORC forcings in the WV case and the NWMv2.1 analysis and assimilation forcing in the FL case. Given its high skillfulness and computational efficiency, we use the 20-member a-EnKF-OI experiment for both domains. In the context of hourly cycling, the ensemble-mean posterior (analysis) at each hour furnishes initial conditions for

predictions spanning up to 18 hours without additional DA. This time frame aligns with the NWMv2.1 short-range forecasts. Given that the same forcings are utilized in the forecasts, our baseline comparison against the OL forecast involves directly assessing the OL run itself.

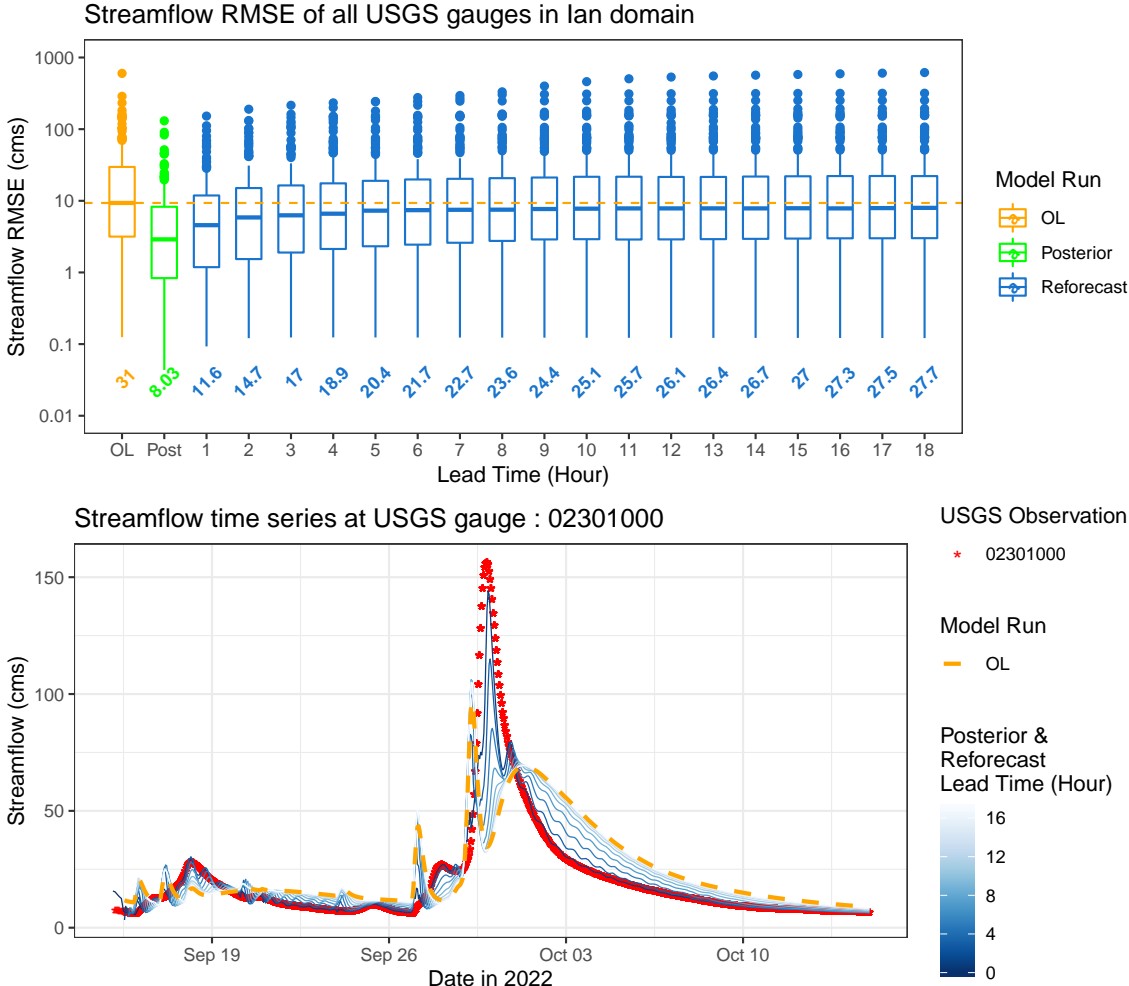

**Figure 15.** Top Panel: Boxplots summarizing the streamflow RMSE of mean ensemble over time for open loop (orange box), posterior (green box), and reforecasts at different lead times (blue boxes). The horizontal dashed line depicts the median of streamflow RMSE for open loop. Averaged RMSE is annotated underneath the individual boxplots. Note that the y-axis is in log-scale. Bottom Panel: Hydrographs for North Prong Alafia River at Keysville FL (Gauge ID: 02301000) during hurricane Ian. Red dots represent the observed streamflow in cms. The orange dashed line depicts the open loop streamflow simulation. The blue lines show the model posterior (lead time of zero) and reforecasts at different lead times.

The skill assessment of reforecasts for the FL flooding case is summarized in the top panel of Figure 15. Across all lead times, the reforecasts consistently demonstrate enhanced performance compared to the open loop. However, this improvement

diminishes with increasing lead time. For example, at an 18-hour lead time, the reforecasted streamflow (averaged over all gauges) initialized by the a-EnKF-OI estimates is 11% more accurate than the OL. To underscore the significance of DA in achieving more accurate forecasts through improved initial states, hydrographs for the North Prong Alafia River at Keysville, FL (Gauge ID: 02301000) are provided in the bottom panel of Figure 15. This gauge, with a drainage area of 135 square miles and experiencing relatively brief flooding, exemplifies notable improvements in the shorter lead times (< 6 hours) compared to the open loop simulation. However, as the lead time extends to 18 hours, the model's response tends to converge toward the OL solution.

Figure 16 encapsulates the reforecast performance for the WV test case. Similar to Fig. 15, the top panel features boxplots illustrating streamflow RMSE across all USGS gauges in the domain for the OL, the posterior, and various reforecast lead times. Just as observed in the FL case, noteworthy enhancements in streamflow predictions during shorter lead times gradually diminish within the first 6 hours. In a median statistical sense, these improvements align closely with the OL forecasts after approximately 10 hours. Interestingly, when comparing FL's Ian test case to the WV test case, forecasts for the Ian case are superior in quality particularly in higher forecast lead times. This distinction arises from the nature of the WV event, characterized as a short-lived flash flood in contrast to the more prolonged event in FL's Ian test case. Consequently, there is relatively less memory of the DA correction in many streams for the WV event compared to the Ian event in FL.

Nevertheless, on average, the reforecasts for the WV test case consistently outperform the OL up to hour 18. This trend is attributed to the fact that the mean RMSE is generally dominated by RMSEs of large rivers, which have an enduring memory, preserve the impact of DA for many hours, resulting in a considerable reduction in their error metrics. Consequently, the mean RMSE across the domain remains lower than that of the OL, even for forecast lead times exceeding 10 hours. The bottom panel in Figure 16 exemplifies this phenomenon at the Kanawha River. In the OL simulation, there is a substantial underestimation of streamflow compared to the observations (depicted by red stars). With the assimilation of streamflow into the a-EnKF-OI, predictions at short lead times are aligned with the observations. However, as the forecast lead time extends, the prediction gradually converges towards the OL solution. Given the nature of this relatively large river and the multi-day duration of the event at this location, the model's estimates remain notably more accurate than the OL even after 18 hours.

## 5   Summary and Discussion

In this study, we have delved into the innovative application of hybrid ensemble and variational data assimilation techniques for streamflow and flood prediction within the WRF-Hydro National Water Model (v2.1) configuration and the Data Assimilation Research Testbed (DART). The resulting "HydroDART" system is specifically tailored to offer precise ensemble streamflow predictions during challenging flood events, such as intense rainfall and hurricanes. HydroDART leverages the ensemble Kalman filter, incorporating adaptive covariance inflation and along-the-stream localization to address issues like sampling errors and bias (e.g., El Gharamti et al., 2021). Our system delivers hourly streamflow analyses utilizing data from the extensive USGS gauging network across the United States.

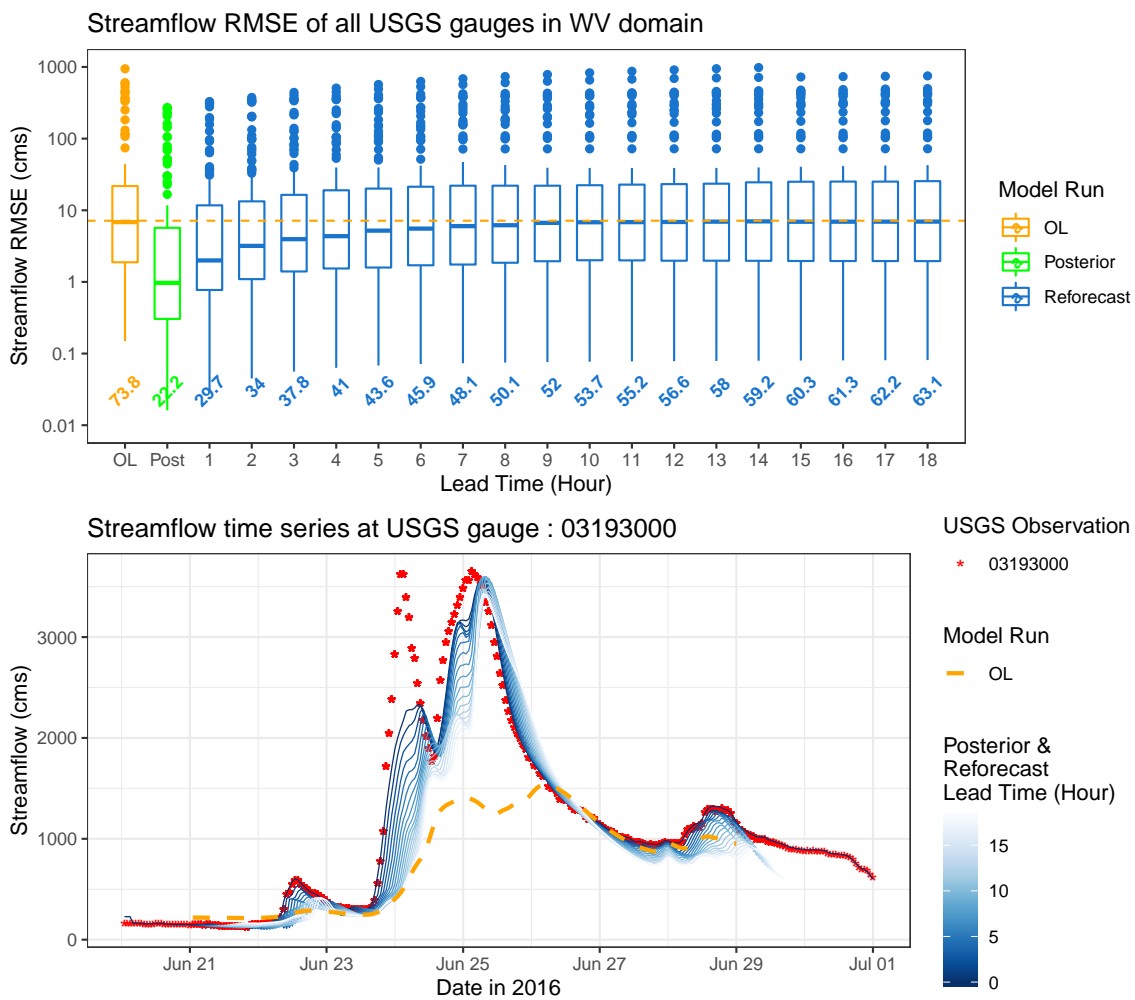

**Figure 16.** Similar to Fig. 15 but for Kanawha River at Kanawha Falls WV (Gauge ID: 03193000).

The hybrid ensemble-variational scheme presented in this paper seamlessly combines the time-varying sample error covariance derived from the ensemble with a static climatological error covariance commonly employed in systems like optimal interpolation and 3/4D-Var. Through extensive testing conducted across two different basins – West Virginia's flash flooding in June 2016 and Florida's inland flooding triggered by Hurricane Ian in August 2022 – we illustrate that the hybrid algorithm enhances the performance of the EnKF, notably improving prediction precision.

Our findings reveal that the judicious blending of the static background covariance with the EnKF effectively improves the ensemble spread, successfully mitigating pronounced model biases observed during flooding events. Optimal results from the hybrid filter were achieved when assigning equal weight to the ensemble and climatology (i.e., a hybrid weight of 0.5). Notably, relying solely on climatological information while disregarding the dynamic ensemble led to degraded results and

yielded poor-quality discharge estimates. Across various gauges in both hydrologic basins, the hybrid scheme exhibited near-perfect alignment with observations, boasting efficiency metrics such as NSE and KGE very close to 1.

Crucially, the hybridized covariance not only heightened prediction skill and reliability but also demonstrated improved efficiency by operating effectively with a time-varying ensemble that utilized only 25% of the members employed by the EnKF alone. Our results indicate that employing 20 realizations in the dynamic ensemble is sufficient to maintain high-accuracy streamflow predictions, consistent with the findings of Abbaszadeh et al. (2019). This, coupled with the NWM submodel design of HydroDART (utilizing only the NWM's streamflow, conceptual groundwater storage, and reservoir models), suggests that a system like HydroDART may be computationally efficient enough for operational use.

Additionally, we explored an adaptive variant of the hybrid scheme that automatically adjusts the hybrid weight at each stream and bucket in the domain based on ensemble statistics. The adaptive scheme, employing 20 members, demonstrated robustness and high skillfulness. Finally, we conducted short-range streamflow forecasts initiated from the hybrid scheme analyses and compared the results to the open loop, revealing consistent improvements in model forecasts for up to 18 hours.

Despite its successful application, the examined framework warrants further research and sensitivity studies. For instance, the ATS localization was specifically tuned for the EnKF and not its hybrid counterparts. It remains conceivable that, with the integration of climatological information, a reduction in localization (i.e., broader cutoff radii) might be possible. The potential for mitigating spurious correlations arising from limited ensemble sizes through the application of the static background covariance was evident in the observations made using the hybrid EnKF-OI scheme, particularly with a minimal number of ensemble realizations in Section 4.3. Furthermore, studying the non-Gaussian aspects of streamflow, as in Hernández and Liang (2018), was not explored in this work. Recent methods such as the Quantile Conserving Ensemble Filtering (QCEF, Anderson, 2022) can be utilized within HydroDART. Using the QCEF, streamflow can be expressed in various non-Gaussian forms during assimilation and this might be a more suitable approach than the on utilized here.

In the case of the adaptive hybrid variant, our study focused on scenarios where the hybrid weight initiates at 0.5. Acknowledging that smaller weights may compromise performance, an exploration of commencing the algorithm with larger weights could be a plausible avenue. This strategy might vary across different domains, considering the dynamic nature of hydrologic basins and water conditions. It is noteworthy that the weight coefficients across the stream network in FL and WV did not converge to specific values. This lack of convergence was attributed to the changing ensemble conditions, such as spread and bias, over our relatively short simulation periods. Extending the simulation duration could potentially lead to the convergence of hybrid weights as streamflow conditions stabilize. In our case, the adaptive algorithm predominantly assigned a balanced weight to the majority of gauges, i.e., $\alpha \in [0.48, 0.52]$. This aligns with the demonstrated optimal performance of a constant homogeneous weight of 0.5 (Section 4.2). For extended simulations, one might consider implementing a reset mechanism for the hybrid weights or exploring the utilization of seasonal climatological covariances rather than a single one.

A potential extension of the present study involves assessing the efficacy of the hybrid DA approach in medium and long-range forecasts. This expansion could encompass a broader array of hydrologic variables, such as stream temperature, and involve additional modeling components like soil moisture and snow. Furthermore, instead of focusing on specific flooding events, our forthcoming investigations will delve into evaluating the performance of the hybrid DA methodology in a compre-

hensive simulation covering the entire CONUS. This strategic shift aims to ascertain the robustness of the latest HydroDART version across diverse hydrologic conditions and to analyze its computational complexity within a large-scale domain.

*Code availability.* The data assimilation code used in this study is openly available as part of the DART repository (directory path: DART/-models/wrf_hydro) on GitHub; https://doi.org/10.5065/D6WQ0202 (last access: 15 September 2022, DART team, 2022). The model code is also freely available and can be accessed at https://ral.ucar.edu/projects/wrf_hydro (last access: 15 September 2022, WRF-Hydro team, 2022).

*Data availability.* The datasets, scripts and routines used to generate the results of this work can be accessed through the following public Zenodo repository: https://doi.org/10.5281/zenodo.5532569 (El Gharamti, 2023).

*Author contributions.* MEG developed the hybrid algorithm and implemented its parallel version in DART, ran some DA experiments and wrote more than 70% of the paper. AR prepared the hydrologic domains, ran the experiments and wrote sections 2.1, 3 and 4.5. JLM maintained the Python configuration framework, developed the retro run procedure in section 3.3 and reviewed the work.

*Competing interests.* The authors declare that they have no conflict of interest.

*Disclaimer.* Any opinions, findings, and conclusions or recommendations expressed in this publication are those of the authors and do not necessarily reflect the views of the National Science Foundation.

*Acknowledgements.* This work was performed as part of NCAR's Short Term Explicit Prediction (STEP) Program, which is supported by the National Science Foundation funds for the United States Weather Research Program (USWRP). The authors would like to acknowledge high-performance computing support from Cheyenne (https://doi.org/10.5065/D6RX99HX) provided by NCAR's Computational and Information Systems Laboratory, sponsored by the National Science Foundation.

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
