# Peer review of "Leveraging a Novel Hybrid Ensemble and Optimal Interpolation Approach for Enhanced Streamflow and Flood Prediction"

_Hydrology and Earth System Sciences, 2023_

## Author Comment (AC1)

**Reviewer #1:**

Summary

In this paper, the authors implement and test a novel data assimilation framework that weights the dynamic component (i.e., time-varying sample error covariance matrix) of the ensemble Kalman Filter (EnKF) and a static component (i.e., a climatology-based covariance matrix) when computing the prior ensemble covariance matrix, with the end goal of improving streamflow simulations and flood forecasts. The framework is implemented in the WRF-Hydro modeling system with the Data Assimilation Research Testbed (DART). The authors conduct a suite of experiments to demonstrate their approach, using two case study flood events occurred in West Virginia (June 2016) and Florida (September 2022). The results presented in the manuscript not only demonstrate the superiority of the proposed method through several verification metrics, but also the computational efficiency, since good evaluation scores can be obtained with only 20 members – as opposed to the 80-member implementation of the EnKF benchmark.

This is a scientifically solid and well written piece of work, with a neat collection of (beautiful) graphics and well-supported conclusions. I commend the authors for the high presentation quality. I have only one main comment that will require some work – though should not be hard to address –, along with several minor comments and editorial suggestions that may be helpful to improve the quality of this manuscript.

We would like to thank the reviewer for their positive review and insightful comments/suggestions. All of the reviewer's comments have been addressed and the changes have been incorporated in the revised manuscript. Please find our detailed response to each individual comment below.

*Main comment*

1. I think that the authors should make an effort to better connect their results with the existing literature (which is nicely reviewed in the introduction). This can be done in a separate section named "Discussion" (after section 4 and before the conclusions section), and a good starting point would be moving all the text in L614-635 to this new section.

   We agree with the reviewer's suggestion. In the revised manuscript, we have made connections between our results and previous studies. Because of the limited hybrid ensemble-variational work on streamflow, we tried our best to include the commonalities with our work in the Discussion section. For instance, similar to the findings in Section 4.3, Abbaszadeh et al. (2019) reported that their PF can be run using a small ensemble size given their hybridized scheme with 4D-VAR. We also

mention our lack of assessment of non-Gaussian streamflow features, something that Hernandez and Liang (2018) explored in their hybrid OPTIMISTS work. We then provide avenues to explore such research using more recent ensemble techniques (e.g., Anderson, 2022).

We also renamed the Conclusion Section to "Summary and Discussion" and decided to keep the discussion of the results and future work together. We hope that this now addresses the reviewer's concern.

*Specific comments*

1. L3: "model deficiencies". Do you mean hydrological model (i.e., structural and parametric) deficiencies?

   Yes, by deficiencies we mean parametric errors such as the presence of unknown/uncertain parameters in addition to other structural errors.

2. L6: to the best of my knowledge, the abstract should not contain citations. Please check the guidelines provided by HESS.

   The reviewer is right. We have checked the HESS guidelines and confirmed that references should not be included in the abstract. Accordingly, we have omitted the citation from the abstract.

3. L9: "validate". I think what you are actually doing is to evaluate the effectiveness of your framework and, therefore, I recommend replacing the word "validation" with "evaluation" throughout the manuscript.

   Thanks for the recommendation. We have replaced the word "validation" with "evaluation."

4. L12: to avoid confusion among readers, please use the terms "significant" or "significantly" only when referring to statistically significant results. In this case, it seems that "considerably" or "substantially" are better options.

   Following the reviewer's suggestion, we have replaced "significantly" by "substantially."

5. L24: In this context, does flooding happen because of streamflow (and hence surface water level) increments? Please clarify.

   Inland flooding refers to the flooding that is caused either by fluvial flooding (overflowing rivers) or pluvial flooding in areas not near a coastline or large body of water (in contrast to coastal flooding). We added the following to the text to clarify.

"Inland flooding could be caused by both the river water level exceeding river bank heights or rainfall intensity exceeding the infiltration capacity. The latter is the major cause of the flooding in case of tropical storms and hurricanes."

6. L30: you might want to refer to "hydrological data assimilation" here, and cite earlier studies on this topic (e.g., Houser et al. 1998; Margulis et al. 2002; Reichle 2008).

The suggested references have been added.

7. L33: I think that it would be more appropriate to cite earlier studies introducing and clarifying the EnKF (e.g., Evensen 1994; Burgers et al. 1998).

Agreed; the earlier EnKF studies are now referenced.

8. L35: what do you mean with "ensemble increments" here? Are you referring to differences between observed and modeled fluxes?

Ensemble increments refer to the added correction during the EnKF update (i.e., increment = analysis - forecast). The difference between observed and modeled fluxes is often called 'innovations.'

9. L38: in my opinion, it is odd to cite McMillan's work (which is amazing) without referring to the retrospective EnKF (Pauwels and De Lannoy 2009, 2006), which inspired the recursive EnKF.

This is a good suggestion. We now mention the work of Pauwels and De Lannoy 2006, 2009 and briefly describe their retrospective EnKF efforts.

10. L44-45: you might want to cite the work of Caleb DeChant when listing particle filter studies (e.g., DeChant and Moradkhani 2011, 2014), and Steven Margulis' particle batch smoother (Margulis et al. 2015).

Thanks for pointing these studies out. We now include them as part of our literature review of the Introduction.

11. L64: since the paper should be self-contained, it would be good adding a concise explanation of "covariance hybridization" here or somewhere else.

Covariance hybridization is described directly in the next paragraph and in Section 2.2.2.

12. L119: I think it would be appropriate to include references for the Noah-MP model (Niu et al. 2011; Yang et al. 2011).

Thanks for pointing out the need for adding the reference. Both references are now added to the paper.

13. L120-122: the explanation about channel, reservoir and conceptual groundwater component are unclear to me. Can you please elaborate and re-word?

Thank you for bringing this to our attention, to make it clear, we made a number changes to the flow of the paragraph and content and change the following paragraph from the original text (L118-128) to the following:

"The NWMv2.1 configuration consists of the Noah-MP land surface model (Niu et al., 2011; Yang et al., 2011), subsurface and surface flow routing, baseflow/groundwater routing, channel and lake/reservoir (i.e., waterbodies) routing (Cosgrove et al. 2024). In each time step, first the land surface model is operated on a coarse resolution of 1 km$^2$. Then, terrain routing (subsurface and surface flow routing) is performed on the 250 m$^2$ grid spacing. NWM utilizes the USGS National Hydrography Data (NHD) Plus Version 2 medium-resolution dataset (McKay et al. 2012), which provides both streams and corresponding catchments. Each stream is represented by a channel/reach vector in the model, and the basin associated with the stream acts as a conceptual groundwater basin/bucket in the model. The inflow to each groundwater bucket/basin is the aggregated outflow from the soil column (1 km LSM grid) to the NHDPlusV2 catchments. Then, a conceptual groundwater routing is performed. The outflow from bucket/basin is estimated based on an exponential storage-discharge function. Next, the outflow from groundwater basin/bucket combined with the lateral channel inflows from the terrain routing are routed through the channels using the Muskingum-Cunge routing method (Read et al. 2023). WRF-Hydro also includes options to represent lakes and reservoirs. Inflow fluxes to lakes and reservoir objects embedded into the NWM routing network are routed using level pool scheme (Gochis et al. 2020, Cosgrove et al. 2024).

Because the channel, reservoir/lake, and conceptual groundwater components are one-way coupled to the other model components, following El Gharamti et al. (2021), a channel, reservoir/lake, and conceptual groundwater submodel of the NWM is used here. This configuration is computationally cheaper compared to the full model and therefore appealing for running an ensemble system. The prognostic variables that are updated in this study are the streamflow discharge and groundwater bucket head. It should be noted that the lake/reservoir objects are defined on the stream reaches, however they are not considered in the state updating."

14. L131 and caption of Figure 1: I suggest replacing the word "forcings" with "input fluxes", since the former is typically used when referring to meteorological forcings in hydrological modeling.

Thank you for pointing out a potential source of confusion. The term forcings has been replaced by "input fluxes" as suggested in two places in the manuscript where it is referring to the input to the channel/bucket component.

15. L139-140: I recommend the authors including a short description of the input ensemble and the channel parameter ensemble in an Appendix. Also, did you calibrate the model error parameters to achieve good statistical properties (i.e., spread, observation indistinguishable from ensemble members) of the open loop ensemble (e.g., Pauwels and De Lannoy 2009; Alvarez-Garreton et al. 2014)?

The perturbations to the input fluxes and the channel parameters have been meticulously tuned for best performance. This was done as part of our previous HydroDART study (El Gharamti et al., 2021). For instance, the channel parameters were sampled according to:

- Geometric quantities ~ U[0.6, 1.4]
- Manning's N parameters ~ U[0.8, 1.8]

under the following conditions: n_cc > 1.5n, T > 1.2B, T_cc > 2T where T is top width, B is the bottom width, n is Manning's N, T_cc is the width of the compound channel, and n_cc is the Manning's N of the compound channel. These choices were made such that enough spread is maintained in the ensemble while integrating the model especially in low flow periods. We also monitored the reliability and the accuracy of the resulting multiphysics streamflow ensemble.

We also tested several forms of distributions to draw the input fluxes including, Gaussians, Gamma, Inverse-Gamma, exponential, etc. For each choice, we selected various parameters that specify the shape of the pdf and compared the results. The best reliability and accuracy was obtained using Gaussian samples with zero mean and 40% flux standard deviation.

All of this was discussed in detail in Section 2 of El Gharamti et al., 2021 and we follow the exact same strategy here. We have briefly added this information in the revised manuscript which reads as follows:

"To perturb the boundary fluxes to the streamflow and bucket models, we use Gaussian samples with zero mean and standard deviation equal to 40% of the flux value at each location. The Gaussian choice yielded the best streamflow estimates (in terms of accuracy and spread consistency) as compared to other tested forms, e.g., gamma, inverse-gamma and exponential. We also perturb the geometric and other channel parameters using uniform noise models. The parameters of the uniform densities were carefully selected such that the resulting streamflow ensembles were found not only skillful but also reliable."

16. L142-143: please add a few sentences describing the parameter estimation process. Note that there are large parameter sensitivities in the Noah-MP model structure (Mendoza et al. 2015b; Cuntz et al. 2016), and their calibration may affect the outcomes of hydrological applications considerably (e.g., Mendoza et al. 2015a).

We have changed the following statement:

"This means that there is a level of model calibration that reduces some of the model background biases, however, there still exists some biases in the model."

To:

"There are a number of parameters in the WRF-Hydro modules, in particular, the NoahMP land surface model (Mendoza et al. 2015; Cuntz et al. 2016, He et al. 2023), with different degrees of sensitivity that could be tuned via calibration. In NWMv2.1, 14 parameters impacting different processes (vegetation, soil, snow, and runoff parameters) were chosen based on the previous sensitivity analysis and studies (Cosgrove et al. 2024). These parameters were calibrated using the iterative Dynamically Dimensioned Search approach (Tolson & Shoemaker, 2007) for a large number of basins throughout the US. The objective function used was one minus weighted Nash–Sutcliffe efficiency (NSE) and logNSE, calculated based on the hourly streamflow simulations. Summary statistics of the model statistics prior and after calibration can be found in Cosgrove et al. (2024). Although the model biases have been reduced through calibration, there still exists some biases in the model."

17. L172: please add the equation for Dy.

Delta y defines the increments in observation space, i.e., the difference between the analysis and predicted ensemble at the observation locations ($Dy = y^a - y^f$). We have added an equation for the Dy in the revised manuscript.

18. L178: If alpha may vary with time, it would be good to add the subscript k.

Indeed alpha will vary in time using the adaptive scheme. A subscript k has been added for the weighting factor in all equations and derivations. Thanks.

19. Figures 2 and 3: please note that not all your readers are familiar with US geography. I suggest merging these into a single figure, adding a panel with a map of the globe that shows the CONUS, and a rectangle showing the geographic extension of the subdomains of Figures 2 and 3. Please add a north arrow and a scale bar to each panel.

Thank you for the insight and suggestions. We have merged Figures 2 and 3, and added a subpanel with the globe which shows the US and the study domains in it. We also added the north arrow and scale to both figures. Please find the revised Figure below.

[Figure]

20. L286-307: Are you selecting a temporal window and extracting years randomly? Since you have only 42 years, I presume you can repeat them to complete 1,000 realizations, right? Perhaps a diagram could help to clarify the procedure.

The explanation of the random draw is explained in lines L299-307 for both cases. Given the length of the manuscript and limited information that could be conveyed through a diagram in this case, we opted to add to the explanation instead. Therefore, the following in bold is added to the text in line 302 as follows:

"Every model simulation within the month of September from these 42-year model simulations is considered a member of the climatology ensemble. **Given the model simulation is at an hourly temporal scale, there are 42 (number of years) * 30 (number of days in September) * 24 (number of hours in a day) = 30,240 realizations to choose from.** Subsequently, 1,000 members are randomly selected from this dataset and preserved for use by HydroDART, as outlined in equation (6).

21. L312: please refer to your performance measures as probabilistic and deterministic verification metrics. I recommend moving all this information to the Methods section (maybe in a table; see Table 2 in Araya et al. 2023), and add it to your methodology diagram.

We would like to thank the reviewer for this good suggestion. We now have moved the verification metrics and their description to section 3. We also adopted Araya

et al. (2023) style and included all the metrics in Table 1, which distinguishes them between deterministic and probabilistic.

22. Related to the previous point, I really think you should add at least one probabilistic verification metric to assess reliability – i.e., adequacy of the simulated/forecast ensemble spread to represent the uncertainty in observations – and report this metric in your DA analysis/comparisons. A good choice would be the α index from the predictive quantile– quantile (QQ) plot (Renard et al. 2010).

We thank the reviewer for bringing this up. The reliability index for the predictive (prior) distribution is now computed and added to all hydrographs (Figs 4, 6, 7, 12, 13). The equation and the reference for RI are also included in Table 1. For instance, this is how Figure 4 looks like in the revised manuscript.

[Figure]

23. Section 4: I suggest renaming this section "Results".

Done.

24. L339-344: all this information should be in the Methods section.

We followed the reviewer's suggestion and moved this text to section 3.4. The new section is titled "Experimental Design and Verification", it reads as follows:

"To test the performance of the hybrid scheme against the EnKF, we perform different assimilation runs in which we set the ATS localization cutoff distance to 100 km and turn on adaptive prior and posterior inflation following El Gharamti et al. (2021).

Our experiments commence in Section \ref{ens_runs} by testing the performance of the EnKF within the HydroDART framework using an ensemble of 80 members. This approach is similar to the experiments outlined in the prior HydroDART study which focused on hurricane Florence in North Carolina (El Gharamti et al., 2021). The objective is twofold: to assess the prediction system's performance in distinct basins characterized by diverse modeling and precipitation complexities, and to establish a baseline performance, both qualitatively and quantitatively, for the

EnKF. This baseline serves as a reference point from which we intend to enhance predictive capabilities through the implementation of the hybrid approach in subsequent sections. For the hybrid EnKF-OI runs, we first examine the sensitivity of the scheme with respect to a few constant choices of the weighting factor (Section 4.2) and explore the impact of hybridizing the background covariance on the ensemble spread and inflation (Section 4.2.1). The idea is to determine whether the inclusion of climatological covariances would nullify the use of inflation in the dynamic ensemble. After determining the optimal hybrid weight (Section 4.2.2), several sensitivity runs with respect to the size of the dynamic ensemble are conducted (Section 4.3). Those runs are aimed to uncover the computational characteristics of the hybrid scheme and figure out whether it can be run more efficiently than the EnKF. The adaptive variant of the hybrid EnKF-OI algorithm is then studied in detail in Section 4.4. Finally, the application of the adaptive scheme for short-range forecasts is investigated in both domains in Section 4.5.

To assess the quality of the estimated streamflow, we use many ensemble and hydrological metrics as shown in Table 1. Some of these metrics are deterministic in nature such as the root mean square error (RMSE) and others are probabilistic such as the reliability index (RI). We provide summary statistics by aggregating a few of these metrics for all flow gauges using boxplots. Where necessary, two sample t-tests are conducted to comment on the statistical significance of one experimental result over others. The metrics are also computed separately for individual hydrographs, low flows and high flows. Low flows are characterized by computing the 10th percentile of observed streamflow at each gauge over the entire assimilation period, while anything above the 90th percentile is deemed a high flow.

From Table \ref{metrics}, the centered root mean square error is used to construct Taylor diagrams (Taylor, 2001) which offer a comprehensive view for all gauges in the present hydrologic domains. For optimal performance, Taylor diagrams are generally characterized by a correlation of 1, with both C-RMSE and standard deviation equal to 0. Rank Histograms (Anderson, 1996) are also utilized to study the reliability of the predicted streamflow ensembles. Flat rank histograms often indicate reliable predictions while skewed ones usually hint to limited ensemble spread and poor coverage of the observation. "

25. Figure 4: why don't you include the KGE and NSE of the posterior estimates? You might want to add letters (a) and (b) to each column (this comment applies to all your figures).

We chose not to add the NSE and KGE metrics for the posterior estimates, in the original submission, because it's always going to be better than the prior. We believe that the priors, i.e., one-hour forecasts, are more appropriate to diagnose

rather than the posteriors which already assimilated the observations. However, following the reviewer's suggestion, we now compute KGE and NSE for the posterior estimates too for consistency. We also followed the reviewer's recommendations and added the letters (a), (b), .. for all subplots.

26. L359-360: How uncertain are the meteorological forcings (in particular, hourly precipitation) used for your model? Perhaps this explains why the hydrological model does not replicate some smaller flood waves.

We have not explored the role of the precipitation uncertainty in the current study in detail. AORC is the atmospheric forcing dataset used for the West Virginia test case. According to Fall et al. 2023, this product has relatively low biases compared to other available products. Although, the uncertainty in the forcing could be very well the reason for some of the errors in streamflow simulations, it is not the only reason. It is hard to decompose which is the main source of the model under performance. For example, the small event on June 21st for gauge 03193000, there is a rainfall event observed in the AORC forcing dataset, however, the model does not produce any streamflow changes. In this case, most likely the infiltrated water is stored as soil moisture and does not produce streamflow response.

27. Figure 5: in the panels with KGE and NSE results, I suggest replacing the y axis title "KGE, NSE" with "metric value" (or something like that) to avoid confusion among readers, since the KGE and NSE are NOT comparable metrics, even if their ranges of possible values are the same (Knoben et al. 2019). I think it would be good to warn readers about this issue somewhere in the text. If the number of values in the boxplots corresponds to the number of stations, it would be good to provide that number in the axis titles or in the figure caption.

We followed the reviewer's suggestion and changed the y-axis label to "metric value." We also mentioned in the revised text that KGE and NSE are not comparable metrics although their ranges are similar. The number of all and reference gauges in each domain has also been added to the plot as shown below. Thank you.

[Figure]

28. Figures 5 and 11: I recommend the authors applying a statistical test to check whether the differences among the empirical probability distributions (i.e., differences between boxplots) are statistically significant. For example, they could apply a two sample t-test to check whether the sample means are statistically different at a specific significance level.

Another good suggestion by the reviewer. A two sample student t-distribution test (with 5% confidence level) has been conducted to investigate the results in Fig. 5 and 11. In the figures below, we show two results: (i) Ian: Open Loop versus HydroDART prior ensemble and (ii) WV: EnKF with 80 members versus EnKF-OI with 20 members. As shown in both cases, the null hypothesis that the RMSE or bias averages are equal has been rejected. In the revised manuscript, we now mention that the results have been analyzed to make sure the reported differences are statistically significant. We include the t-test results for Ian in Figure 11.

[Figure]

[Figure]

[Figure]

[Figure]

29. L374-380: this description should be in the Methods section.

    Part of the text has been moved to the new section 3.4. We revised the introductory text in Section 4.2 to enhance the flow of the results.

30. Figure 6: in my opinion, there is no need to repeat the entire legend in all panels.

    We appreciate the reviewer's opinion, however, we believe having the legend for all hydrographs is necessary because of metrics such as the observation rejection and the RMSE. The observation rejection changes depending on the efficiency of

the underlying ensemble technique. Also, this makes the plotting consistent for other figures where only 2 panels are available.

31. L388: "ensemble uncertainty". Please refer to ensemble spread throughout the paper, since the true uncertainty in your systems is unknown.

The term uncertainty has been replaced with spread. We agree with the reviewer's assessment.

32. L412-413: I suggest moving this text to the methods section and explaining why is it worth analyzing this.

This is now moved to section 3.4.

33. L415-416: it would be good adding two panels at the top of Figure 9, with precipitation and streamflow time series to visualize this.

Streamflow and precipitation time series have been to Fig. 9 as recommended. Here is the modified figure:

[Figure]

34. L425-430: this information should be in a section dedicated to verification metrics.

As suggested, this information is now moved to Section 3.4.

35. L438-439: this text should be at the beginning of subsection 4.3.

Done.

36. Table 1: I don't think you need more than three decimals.

Agreed, we followed the reviewer's suggestion and only kept 3 decimals.

37. L441-444: please move this text to the methods section.

Done.

38. Figure 10: It is really difficult to differentiate among dots in the top panels. If you want to highlight that EnKF is the worst DA strategy, I recommend changing the symbol type (perhaps replace dots by crosses?). Similarly, you could modify the symbols for other configuration standing out.

We changed the markers for the different runs to make the results more visible.

39. L475, L534, L535 and everywhere else: despite this is a matter of style, I recommend deleting bombastic adjectives (e.g., "exceptional", "remarkable") and showing your numbers instead. Let the readers judge your proposed approach.

We have reduced the usage of these adjectives as suggested.

40. L498-499: this is really hard to see because the dots in Figure 14 are too small. Please consider increasing their size.

Thanks for pointing this out. We have increased the size of the makers showing the location of the gauges (also changed the maker style to make it easy to visualize).

41. L503-504: this is very hard to visualize. Why don't you just show a scatter plot between the weights and the distance to the landfall to make the point?

We're sorry this was hard to visualize. In the revised manuscript, the colors in the figure have been enhanced and the text now clearly distinguishes between the reaches which have undergone smaller (gray colored) versus larger (green/pink colored) changes in the weighting coefficients.

42. L518-519: please move this text to the methods section.

Done. The description of low and high flows is now included in Section 3.4.

43. L538-554: please move this text to the methods section.

For this part of the text, we decided to keep it in Section 4.5 which can be seen partially as discussion of the results and further examination. A brief explanation has been provided in Section 3.4 though.

44. L555-558: please move this description to the caption of Figure 16.

Done.

45. L558-559: "the EnKF effectively improves the ensemble spread". I think that this statement is unsupported, unless you include QQ plots or rank histograms to assess how adequate is the spread relative to the observations.

In order to support our statement, we have included rank histograms for different gauges in both domains. They are now appended to Figure 8 of the revised manuscript (and also shown here).

[Figure]

[Figure]

[Figure]

As can be seen, the hybrid scheme is able to provide relatively flat rank histograms at both locations. In panel (c), it's clear how the hybrid scheme improves the reliability of the ensemble unlike the EnKF which tends to overestimate the observed flow (RH is skewed to the right). A similar discussion has been added to the revised manuscript.

*Suggested edits*

1. L1-2: "accurately predicting" -> "the accurate prediction of". Done
2. L10: "test cases" -> "case studies". Done
3. L83: "streamflow flooding problems" -> "streamflow forecasting" or "flood forecasting" problems. Done
4. L95: "September 15 to October 15, 2002". Move to the end of the sentence, maybe in parentheses. Done
5. L97 and L265: "hourly" -> "at hourly time steps". Done
6. L118: I think a word is missing between "2.1" and "standard". Maybe "including"? This is how the term is used by NOAA to refer to the NWM.

7. L126: "exceed" -> "exceeds". Done
8. L363: delete "clearly". Done
9. L576: "…of large rivers. Large rivers…" ->   "...of large rivers, which have an enduring memory..." Done

---

## Author Comment (AC2)

**Reviewer #2:**

General comments

This paper presents an innovative application of a hybrid data assimilation algorithm, EnKF-OI (Optimal Interpolation), for streamflow and flood prediction. The hybrid algorithm, developed by El Gharamti et al. (2021), offers novel advancements in hydrologic prediction. It enhances the ensemble spread of EnKF through two key mechanisms: (1) incorporation of time-invariant climatological error covariance into the prior ensemble covariance matrix, and (2) integration of along-the-stream localization. The study conducts a comprehensive evaluation of the hybrid algorithm using two case studies: flash floods in West Virginia and long-term flooding in Florida. Results are analyzed across four main dimensions: (1) weighting between dynamic and static covariances, (2) dynamic ensemble size, (3) adaptive weight adjustment, and (4) short-term streamflow forecasts. The hybrid data assimilation algorithm shows promising performance in two applications.

The paper fits the scope of the HESS. The innovation of this research is clear to me. The experiment design, result analysis, and the presentation of this paper are of good quality. It is a pleasure of reading this research. I suggest a minor revision.

We would like to thank the reviewer for their positive and constructive review. All reviewer's comments have been addressed. Please find our response below.

Minor comments to the authors

1.      Line 6: Consider adding a reference to "El Gharamti et al." in the abstract.

We have checked the HESS guidelines for adding citations to the abstract and found out that it's not allowed, so we remove the reference.

2.      Line 44: Ensure consistency in abbreviating "United States" (USA or US).

We decided to stick to the US and be consistent as suggested by the reviewer.

3.      Line 85: Clarify the difference/relationship/innovation between the existing HydroDART system and the EnKF-OI algorithm employed in this study. Is the EnKF-OI algorithm newly developed or already in the HydroDART system? This clarification would highlight any methodological innovation in the paper.

We would like to thank the reviewer for pointing this out. Indeed, for this work we have implemented the adaptive hybrid EnKF-OI scheme in DART from scratch. This marks the inaugural implementation of hybrid ensemble-variational filters in DART. It's also the first realistic large-scale application of the adaptive hybrid scheme. The added text reads as follows:

"The implementation of the hybrid ensemble-variational scheme is the first of its kind within DART and features several flavors for updating the hybrid weighting coefficients including: constant weights, time-varying homogeneous weights in addition to the more comprehensive temporally and spatially varying weights (as in this work)."

4.  Line 118: Provide a brief explanation of "nudging."

Thank you for pointing out the need to expand, in response this sentence has been added.

"Streamflow nudging is the current data assimilation methodology in NWM operationally. "Nudging" also known as direct insertion refers to moving the modeled flow towards the observed discharge at each time step of the routing model."

5.  Lines 125-127: Consider rephrasing this sentence for clarity or conduct a grammar check.

Thanks for the suggestion! This section has been restructured based on the input from both reviewers.

6.  Line 202: Confirm if the sentence "The notation… is equivalent to the trace of matrix A" is used in the preceding equation.

The equation has been corrected to reflect the variables in the preceding equation.

7.  Line 206: Check the format of the reference "El Gharamti (2021)."

Done.

8.  Lines 397-398: Clarify if this sentence refers to the last subplot of Figure 7.

We added text clarifying that the discussion is referring to the bottom panels of Fig. 7.

9.  Is it possible to shorten this paper by moving some results (e.g., $2_{nd}$ case study relevant content) to the supplementary? The current manuscript is relatively long.

We understand the reviewer's concern, however, the choice to test the adaptive hybrid EnKF-OI scheme in two different domains was intentional. We wanted to validate and test the robustness of the scheme in varying hydrological conditions; in this case (i) quick flash flooding and (ii) long lasting inland flooding due to hurricanes. We also tried to present a detailed verification process in order to cover all aspects of the algorithms and uncover its characteristics under several conditions: e.g., low flow, high flow, bias, reliability, accuracy, etc. We strongly believe that having both test cases in the manuscript is important to support our conclusions.

---

## Author Response (AR2)

I want to thank the authors for thoroughly addressing the suggestions that I provided on an earlier version of this manuscript. Despite it has been improved considerably, I have a suite of specific comments that I would like the authors to address before this paper is accepted for publication.

*We would like to thank the reviewer for their great contributions that improved the quality of the work. We appreciate their time and effort while reading our work. Please find detailed response to all raised comments and questions below.*

1. L3: the authors clarified that they are referring to structural and parametric model uncertainties with "model deficiencies" in the response document, though not in the revised manuscript. Please include that clarification in the abstract.

*Good catch! The clarification has been included in the revised manuscript which reads as follows: "Traditional data assimilation methods face challenges during extreme rainfall events due to numerous sources of error, including structural and parametric model uncertainties, forcing biases and noisy observations."*

2. L35: the authors clarified what they mean with "ensemble increments" in the response document, but not in the revised manuscript. Please include that clarification here.

*The ensemble increments have been clarified as follows: "The EnKF employs a minimum variance estimator, utilizing observations to compute ensemble increments (i.e., difference between analysis and forecast) that are subsequently linearly combined with the predicted ensemble."*

3. L151, L185 and everywhere else: I do not think that a parameter perturbation approach is the same as a multiphysics approach. In order to achieve the latter, they should have used multiple model structures, which is indeed possible with Noah-MP and other modular modeling platforms like FUSE (Clark et al. 2008), SUMMA (Clark et al. 2015a,b), MARRMoT (Knoben et al. 2019), Raven (Craig et al. 2020), etc. Although I did not catch this in my first review, I think this is a critical point that needs to be revised to avoid confusion among readers.

*In the revised manuscript, we omitted the use of the term multiphysics. Both occurrences have been modified accordingly. For instance, L151 now reads: "In addition to this time-varying uncertainty, we also generate an invariant ensemble of channel parameters similar to the configuration of El Gharamti et al. (2021)."*

4. L153-154: do you mean that increasing the variability in the ensemble can help your DA scheme to effectively estimate hydrological model states. If yes, please rewrite the text to reflect this.

*The text has been rephrased as follows: "Because ensemble DA depends on probabilistic forecasts, enhancing the variability within the ensemble can aid the method in accurately estimating the states of the hydrological model."*

5. L168-169: I recommend adding the calibration objective function as an equation in the paper. This is relevant information for the sake of reproducibility.

*We have added the objective function and edited the text of the revised manuscript as suggested: "The objective function was one minus weighted Nash-Sutcliffe efficiency (NSE, Nash and Sutcliffe, 1970) and NSE of logarithmic streamflow (NSElog), both calculated based on the hourly streamflow simulations."*

$$\text{Minimize } J = 1 - 0.5(NSE + NSElog).$$

6. Equation 6: can the authors please clarify if (and how) this equation relates to the calculation of the model error covariance matrix P, and the observation error covariance matrix R?

*In DART, the observations are assimilated serially and thus the full state and observation covariance matrices P and R are never constructed. This makes the computations a lot easier because matrix operations (such as SVD, inverse, etc) are not needed. The serial formulation of the ensemble Kalman filter in DART is separated into 2 steps: update in observation space and then regression onto state space. R (a diagonal element for a single observation) for instance is used to compute the observation increments in equation (5). The covariance sigma_{xy} in equation (6) is one entry of PH^T corresponding to the covariance between observation y and the j^{th} element of the state (H is the observation operator). The details of this serial algorithm have been covered extensively in the literature, starting with the original work of Anderson (2003) which is cited right before equation (5).*

*We added text addressing the observation error: "the EnKF solution is obtained as a linear regression of the observation increments Dy on the entire state vector. We note that the assimilated observations are noisy with Gaussian errors and their observation error covariance is accounted for when computing the observation increments in (5)."*

*We also added a clarifying text at the end of Section 2.2.1 and it reads as follows: "In terms of implementation, we note that the full state covariance in eq. (6) is never constructed using this 2-step serial update scheme. This also applies for the observation error covariance matrix assuming that the observations are uncorrelated in space. For more details on the algorithm and its implementation, the reader is referred to the work of Anderson (2003)."*

7. Figure 2: it is quite odd having panel (a) on the right and (b) on the left. Please consider switching the order of those panels.

*The panels on Figure 2 have been switched as suggested.*

8. L304, 374, 416, 422, 605, 621 and everywhere else in the manuscript: please revise the use of the word "significantly". This is a point I raised in my first review, and I think that needs to be addressed more carefully.

*Although we don't see any problem with the use of the word significantly, we decided to re-word as suggested. Please find the changes below:*

*L304: "It helps avoid assimilating inaccurate observations, and it prevents the inclusion of observations where the mean of the ensemble members is quite far from the observation value."*
*L374: "both the prior and posterior ensemble estimates exhibit improved accuracy"*
*L416: "This area was affected severely by the flooding event under consideration."*
*L422: "although it still shows superior performance compared to the EnKF (bottom panels)."*
*L605: "predictions at short lead times are aligned with the observations."*
*L621: "we illustrate that the hybrid algorithm enhances the performance of the EnKF, notably improving prediction precision."*

9. L393: please re-word this sentence to make it clear that NSE and KGE share the same range of variation.

*The sentence has been re-worded as follows: "It's important to emphasize that the KGE and NSE are not directly comparable metrics, despite having the same range of variation."*

10. L422-423: Are the authors referring to the shift in performance metrics between panels (a) and (b)? Note that panel (b) contains results for alpha = 0.1 (i.e., 10% weight to the dynamic component of the covariance, according to equation 7). Then why did the authors write that "outstanding performance can be achieved by incorporating only 10% of the hybrid covariance from the climatology"? Should it be the opposite (i.e., 10% from the dynamic                                                                                                     component)?
Related to this point, the authors refer to "The introduction of climatological information with alpha = 0.1" in L434. However, according to equation (7) alpha = 0.1 means that you are giving a 10% weight to the dynamic component, and 90% to the climatological component (i.e., alpha = 0.1 means that the authors are introducing dynamic information and not the other way around). Based on this, I strongly recommend revisiting this interpretation of weights and/or correct equation (7).

*Regarding the reviewer's first question, we're referring to panel (f) in which alpha is 0.9 and so 10% (1-0.9=0.1) of the background covariance come from climatology. This is now clarified in the revised text as follows: "Remarkably, such performance can be achieved by incorporating only 10% (i.e., alpha = 0.9) of the hybrid covariance from the climatology in panel (f)."*

*As for the second comment, we think the use of the word introduction is confusing. We meant to say, the first-time climatological information is incorporated, and so eq. (7) is correct. To avoid confusion, we have revised the text as follows: "For alpha = 0.1, most of the weight is placed on the climatological information resulting in a notable increase in the ensemble spread."*

11. L442: the authors refer to an "overestimation", though it seems that there is underestimation because a large fraction of observations (nearly 40%) is larger than the simulated ensemble members according to the rank histogram. Please revise and re-word if needed.
*The sentence has been revised as follows: "At Cowpasture River in the second domain (panel c), a large fraction of the observations (nearly 40%) appears to be larger than the simulated discharge indicating underestimation. The rank histogram of the EnKF also shows partial skewness to the right."*

12. The authors refer to an "improved estimate of the uncertainty", though what they are getting is an improved ensemble spread based on the relative range of variation of the observations.
*The term uncertainty is replaced with ensemble spread as follows: "The hybrid scheme successfully mitigates that bias and yields an improved ensemble spread."*

13. L457-458: This is still VERY hard to visualize from Figure 9, especially in the left panel. Consider decreasing the size of symbols for the hybrid configurations.
*We have followed the reviewer's suggestion and decreased the marker size of the hybrid runs. Please find the updated figure in the revised manuscript. We hope that the figure is clearer now.*

14. L461: the authors state that "The EnKF-OI schemes yield comparable correlations, with alpha = 0.5 consistently offering the best performance in both domains". I do not think this is true for FL, where alpha = 0.7 offers the highest correlation value (according to Table 2).
*The text is now revised, and it reads as follows: "The EnKF-OI schemes yield comparable correlations, with alpha= 0.7 and 0.5 offering the best performance in FL and WV,*

*respectively."*

15. Figure 13: all the descriptions provided by the authors regarding this figure are still very hard to visualize. Because of this and the length of the manuscript (which contains a tremendous amount of information), I would consider removing these results or sending them to supplementary material. In any case, the authors should make the final choice on this matter.

*We thank the reviewer for their suggestion. We really do believe that the spatial variations of the hybrid weight (and the connection to inflation) are an integral part of the story of this manuscript. As such, we would like to keep Fig. 13 and the associated discussion as part of the article.*

16. L556-558: please revise this sentence, because I do not see the consistent improvement that the authors describe when comparing red and blue boxplots, especially in panel (a).

*Thanks for pointing this out. We have revised the test to reflect the results in Figure 14 as follows: "Compared to the 20-member EnKF-OI with a fixed alpha, the adaptive variant demonstrates relatively similar accuracy with a slight advantage in WV's low-flow diagnostics."*

17. L558: what is an outstanding score?

*Here is the definition of the word outstanding from Cambridge Dictionary:*
*"outstanding: clearly very much better than what is usual."*
*So, in this context the EnKF-OI with 80 members has a clear superior performance compared to the other schemes on the figure. While we don't see any issue with the term, we have removed it based on the reviewer's comments.*

18. L141: "using level pool scheme" -> "using a level pool scheme". *Done*
19. L142-143: this sentence reads repetitive. Maybe just write "we use a channel, reservoir, and conceptual groundwater submodel of the NWM, following...". *Done*
20. L157-158: please place "distributions, e.g., gamma, inverse-gamma and exponential" between parentheses. *Done*
21. Caption of Figure 1: "Dotted box" -> "The dotted box". *Done*
22. L170: "Summary statistics of the model statistics" -> "Summary model statistics". *Done*
23. L170-171: awkward sentence. Maybe rewrite as "It should be noted that some model biases remain after the calibration process". *Done*
24. L422 and everywhere else: please delete the word "outstanding". There is no need to use bombastic adjectives. *Done*
25. 565: "until now" -> "so far". *Done*